# Kernelized Cumulants: Beyond Kernel Mean Embeddings

**Patric Bonnier**[1*]     **Harald Oberhauser** [1]     **Zoltán Szabó**[2]

[1]Mathematical Institute, University of Oxford     [2]Department of Statistics, London School of Economics

`bonnier,oberhauser@maths.ox.ac.uk`
`z.szabo@lse.ac.uk`

## Abstract

In $\mathbb{R}^d$, it is well-known that cumulants provide an alternative to moments that can achieve the same goals with numerous benefits such as lower variance estimators. In this paper we extend cumulants to reproducing kernel Hilbert spaces (RKHS) using tools from tensor algebras and show that they are computationally tractable by a kernel trick. These kernelized cumulants provide a new set of all-purpose statistics; the classical maximum mean discrepancy and Hilbert-Schmidt independence criterion arise as the degree one objects in our general construction. We argue both theoretically and empirically (on synthetic, environmental, and traffic data analysis) that going beyond degree one has several advantages and can be achieved with the same computational complexity and minimal overhead in our experiments.

**Keywords:**   kernel, cumulant, mean embedding, Hilbert-Schmidt independence criterion, kernel Lancaster interaction, kernel Streitberg interaction, maximum mean discrepancy, maximum variance discrepancy

## 1   Introduction

The moments of a random variable are arguably the most popular all-purpose statistic. However, cumulants are often more favorable statistics than moments. For example, if $\mu_m := \mathbb{E}[X^m]$ denotes the moments of a real-valued random variable $X$, then $\mu_2 = \mu_1^2 + \text{Var}(X)$ and hence the variance that directly measures the fluctuation around the mean is a much better statistic for scale than the second moment $\mu_2$, see Appendix A. Cumulants provide a systematic way to only record the parts of the moment sequence that are not already captured by lower-order moments. While the moment and cumulant sequences $(\mu_m)_m$ and $(\kappa_m)_m$ carry the same information, cumulants have several desirable properties that generalize to $\mathbb{R}^d$-valued random variables (McCullagh, 2018). Among these properties of cumulants, the ones that are important for our paper is that they can characterize distributions and statistical (in)dependence.

**Kernel embeddings.**   Mean and covariance arise naturally in the context of kernel-enriched domains. Kernel techniques (Schölkopf and Smola, 2002; Steinwart and Christmann, 2008; Saitoh and Sawano, 2016) provide a principled and powerful approach for lifting data points to a so-called reproducing kernel Hilbert space (RKHS; Aronszajn 1950). Considering the mean of this feature – referred to as kernel mean embedding (KME; Berlinet and Thomas-Agnan 2004; Smola et al. 2007) – also enables one to represent probability measures, and to induce a semi-metric referred to as maximum mean discrepancy (MMD; Smola et al. 2007; Gretton et al. 2012) and an independence measure called the

Hilbert-Schmidt independence criterion (HSIC[1]; Gretton et al. 2005). It is known that MMD is a metric when the underlying kernel is *characteristic* (Fukumizu et al., 2008; Sriperumbudur et al., 2010). HSIC captures independence for $d = 2$ components with characteristic kernels (Lyons, 2013, Theorem 3.11) and for $d > 2$ components (Quadrianto et al., 2009; Sejdinovic et al., 2013a; Pfister et al., 2018) with universal ones (Szabó and Sriperumbudur, 2018). MMD belongs to the family of integral probability metrics (IPM; Zolotarev 1983; Müller 1997) when in the IPM the underlying function class is chosen to be the unit ball of the RKHS. MMD and HSIC (with $d = 2$) are known to be equivalent (Sejdinovic et al., 2013b) to the notions of energy distance (Baringhaus and Franz, 2004; Székely and Rizzo, 2004, 2005)—also called N-distance (Zinger et al., 1992; Klebanov, 2005)– and distance covariance (Székely et al., 2007; Székely and Rizzo, 2009; Lyons, 2013) of the statistics literature. Both MMD and HSIC can be expressed in terms of expectations of kernel values which can be leveraged to design efficient estimators for them; we will refer to this trick as the *expected kernel* trick. A recent survey on mean embedding and their applications is given by Muandet et al. (2017). The closest previous work to ours is by Makigusa (2020) who considers the variance in the RKHS – which in our setting can be identified as the first kernelized cumulant after the kernel mean embedding – for two-sample testing like we do in parts of this paper. Unfortunately, Makigusa (2020) does not provide conditions on the validity of the resulting maximum variance discrepancy, and it is not formulated in the context of cumulant embeddings.

**Contribution.** The main contribution of our paper is to introduce cumulants of random variables in RKHSs and to show that under mild conditions the proposed kernelized cumulants characterize distributions (Theorem 2) and independence (Theorem 3). Thanks to the RKHS formulation, kernelized cumulants have computable estimators (Lemma 2 and Lemma 3) and they show strong performance in two-sample and independence testing on various benchmarks (Section 4). Although cumulants are a classic tool in multi-variate statistics, they have not received attention in the kernel literature. The primary technical challenge to circumvent in the derivation of fundamental properties of cumulants is the rich combinatorial structure which already arises in $\mathbb{R}^d$ from their definition via moment-generating function which is closely linked to the partition lattice (Speed, 1983, 1984). In an RKHS, even the definition of cumulants is non-straightforward. The key insight for our extension is that the combinatorial expressions for cumulants in $\mathbb{R}^d$ can be generalized by using tools from tensor algebras. This in turn allows us to derive the main properties of the RKHS cumulants that underpin their statistical properties.

**Broader impact & limitations.** We do not see any direct negative societal impact arising from the proposed new set of all-purpose kernel-based divergence and dependence measure. Choosing the underlying kernels in an optimal fashion—even for MMD in two-sample testing (Hagrass et al., 2022) or goodness-of-fit testing (Hagrass et al., 2023)—and showing optimal rates—even for MMD with radial kernels on $\mathbb{R}^d$ (Tolstikhin et al., 2016)— are quite challenging problems requiring dedicated analysis, and are not addressed here.

**Outline.** The paper is structured as follows: In Section 2 we formulate the notion of cumulants of random variables in Hilbert spaces. In Section 3 we prove a kernelized version of the classical result of $\mathbb{R}^d$-valued random variables on the characterization of distributions and independence using cumulants. We show that one can leverage the expected kernel trick to derive efficient estimators for our novel statistics, with MMD and HSIC arising specifically as the "degree 1" objects. In Section 4 we demonstrate numerically that going beyond degree 1 is advantageous. We provide a technical background (on cumulants, tensor products and tensor algebras), proofs, further details on numerical experiments, and our V-statistic based estimators in the Appendices.

## 2 Moments and cumulants

We briefly revisit classical cumulants and define cumulants of random variables in Hilbert spaces.

---

[1]HSIC is MMD with the tensor product kernel evaluated on the joint distribution and the product of the marginals, or equivalently HSIC equals to the Hilbert-Schmidt norm of the cross-covariance operator.

**Moments.** Let $\gamma$ be a probability measure on $\mathbb{R}^d$ and $(X_1, \ldots, X_d) \sim \gamma$. The moments $\mu(\gamma) = (\mu^{\mathbf{i}}(\gamma))_{\mathbf{i} \in \mathbb{N}^d}$ of $\gamma$ are defined as

$$\mu^{\mathbf{i}}(\gamma) := \mathbb{E}\left[X_1^{i_1} \cdots X_d^{i_d}\right] \in \mathbb{R}, \tag{1}$$

where $\mathbf{i} = (i_1, \ldots, i_d) \in \mathbb{N}^d$ denotes a $d$-tuple of non-negative integers $(i_1, \ldots, i_d \geq 0)$. The *degree* of an element $\mathbf{i} \in \mathbb{N}^d$ is defined as $\deg(\mathbf{i}) := i_1 + \cdots + i_d$. For $m \in \mathbb{N}$, let $\mu^m(\gamma) := (\mu^{\mathbf{i}}(\gamma))_{\deg(\mathbf{i})=m}$ which we refer to as the $m$-th moments of $\gamma$ with the convention that $\mu^0(\gamma) = 1$.

**Cumulants.** Cumulants $\kappa(\gamma) = (\kappa^{\mathbf{i}}(\gamma))_{\mathbf{i} \in \mathbb{N}^d}$ can be defined by the moment generating function as

$$\sum_{\mathbf{i} \in \mathbb{N}^d} \kappa^{\mathbf{i}}(\gamma) \frac{\boldsymbol{\theta}^{\mathbf{i}}}{\mathbf{i}!} = \log \sum_{\mathbf{i} \in \mathbb{N}^d} \mu^{\mathbf{i}}(\gamma) \frac{\boldsymbol{\theta}^{\mathbf{i}}}{\mathbf{i}!}, \ \boldsymbol{\theta} = (\theta_1, \ldots, \theta_d) \in \mathbb{R}^d, \tag{2}$$

where we denote $\mathbf{i}! = i_1! \cdots i_d!$ and $\boldsymbol{\theta}^{\mathbf{i}} = \theta_1^{i_1} \cdots \theta_d^{i_d}$; an equivalent definition of cumulants is via a combinatorial expression of partitions (elaborated in Appendix C.1). Cumulants have several attractive properties, the following forms our main motivation.

**Theorem 1** (Characterization of distributions with cumulants on $\mathbb{R}^d$, from Proposition 1 in Jammalamadaka et al. 2006). *Let $\gamma$ be a probability measure on a bounded subset of $\mathbb{R}^d$ with cumulants $\kappa(\gamma)$ and let $(X_1, \ldots, X_d) \sim \gamma$. Then*

*1. $\gamma \mapsto \kappa(\gamma)$ is injective.*

*2. $X_1, \ldots, X_d$ are jointly independent if and only if $\kappa^{\mathbf{i}}(\gamma) = 0$ for all $d$-tuples of positive integers $\mathbf{i} \in \mathbb{N}_+^d$.*

## 2.1 Moments in Hilbert spaces

Instead of directly considering the law of a tuple of random variables $(X_1, \ldots, X_d)$ in a product space $\mathcal{X}_1 \times \cdots \times \mathcal{X}_d$, it can be advantageous to use feature maps $\Phi_i : \mathcal{X}_i \to \mathcal{H}_i$ and instead study the distribution of the $\mathcal{H}_1 \times \cdots \times \mathcal{H}_d$-valued random variable $(\Phi_1(X_1), \ldots, \Phi_d(X_d))$. Motivated by this lifting, we study here moments of Hilbert-space valued random variables and assume in this subsection (with a slight abuse of notations) that one has already applied the lifting and $X_i \in \mathcal{H}_i$ where $i = 1, \ldots, d$. In Section 3 we specialize the construction to RKHSs, and use these moments (Def. 1) to define kernelized cumulants.

**Moments.** In the finite-dimensional case (1) we defined the moment sequence by taking expectations of products of the coordinates of the underlying random variable. For the infinite-dimensional case, it is convenient to develop a coordinate-free definition which can be accomplished by using tensors. To do so we make use of the following results about Hilbert spaces: for real Hilbert spaces $\mathcal{H}_1$ and $\mathcal{H}_2$ the tensor product $\mathcal{H}_1 \otimes \mathcal{H}_2$ is the Hilbert space given by completion of the tensor product of $\mathcal{H}_1$ and $\mathcal{H}_2$ as vector space; we also write $\mathcal{H}_1^{\otimes m} := \underbrace{\mathcal{H}_1 \otimes \cdots \otimes \mathcal{H}_1}_{m\text{-times}}$. Similarly, the direct sum

$\mathcal{H}_1 \oplus \mathcal{H}_2$ is a Hilbert space. It is natural to consider $\mathbb{E}\left[X_1^{\otimes m}\right] \in \mathcal{H}_1^{\otimes m}$ as the $m$-th moment of a $\mathcal{H}_1$-valued random variable $X_1$ where the integral in the expectation is meant in Bochner sense. Consequently the natural state space for all moments of a $\mathcal{H}_1$-valued random variable is the tensor algebra $\mathrm{T}_1 := \prod_{m \geq 0} \mathcal{H}_1^{\otimes m}$ where by convention $\mathcal{H}_1^{\otimes 0} := \mathbb{R}$. See Appendix B for more details on tensor products of Hilbert spaces and tensor algebras.

**Example 2.1** ($\mathcal{H}_1 = \mathbb{R}^d$, $m = 2$). *If $X_1 = (X_1^1, \ldots, X_1^d)$ is $\mathcal{H}_1 = \mathbb{R}^d$-valued then $\mathbb{E}\left[X_1^{\otimes 2}\right] \in (\mathbb{R}^d)^{\otimes 2}$ can be identified with a $(d \times d)$-sized matrix whose $(i,j)$-th entry is $\mathbb{E}\left[X_1^i X_1^j\right]$.*

Since we are interested in the general case of a $\mathcal{H}_1 \times \cdots \times \mathcal{H}_d$-valued random variable $X = (X_1, \ldots, X_d)$ we arrive at the definition below.

**Definition 1** (Moments in Hilbert spaces). *Let $\gamma$ be a probability measure on $\mathcal{H} := \mathcal{H}_1 \times \cdots \times \mathcal{H}_d$ and let $(X_1, \ldots, X_d) \sim \gamma$. We define*

$$\mu^{\mathbf{i}}(\gamma) := \mathbb{E}[X_1^{\otimes i_1} \otimes \cdots \otimes X_d^{\otimes i_d}] \in \mathcal{H}^{\otimes \mathbf{i}}, \quad \mathcal{H}^{\otimes \mathbf{i}} := \mathcal{H}_1^{\otimes i_1} \otimes \cdots \otimes \mathcal{H}_d^{\otimes i_d} \tag{3}$$

*for every* $\mathbf{i} \in \mathbb{N}^d$ *whenever the above expectation exists. The moment sequence is defined as the element*

$$\mu(\gamma) = (\mu^{\mathbf{i}}(\gamma))_{\mathbf{i} \in \mathbb{N}^d} \in \mathrm{T} := \mathrm{T}_1 \otimes \cdots \otimes \mathrm{T}_d, \text{ with } \mathrm{T}_j := \prod_{m \geq 0} \mathcal{H}_j^{\otimes m},$$

*and for* $m \in \mathbb{N}$ *we refer to* $\mu^m(\gamma) = \bigoplus_{\mathbf{i} \in \mathbb{N}^d : \deg(\mathbf{i}) = m} \mu^{\mathbf{i}}(\gamma)$ *as the* $m$-*moments of* $\gamma$.

In case of $\mathcal{H}_i = \mathbb{R}$, both definitions (1) and (3) apply for $\mu^{\mathbf{i}}(\gamma)$. Henceforth, we always refer to (3) when we write $\mu^{\mathbf{i}}(\gamma)$. Even in the finite-dimensional case, Def. 1 is useful, for instance when $X_1 \in \mathcal{H}_1$ and $X_2 \in \mathcal{H}_2$ have different state space ($\mathcal{H}_1 \neq \mathcal{H}_2$).

## 3   Kernelized cumulants

We lift a random variable $X = (X_1, \ldots, X_d) \in \mathcal{X} = \mathcal{X}_1 \times \cdots \times \mathcal{X}_d$ via a feature map $\Phi : \mathcal{X} \to \mathcal{H}$ into a Hilbert space valued random variable $\Phi(X)$. For the rest of the paper (i) $\mathcal{X}_1, \ldots, \mathcal{X}_d$ will denote a collection of Polish spaces, but the reader is invited to think of them as finite-dimensional Euclidean spaces, (ii) $\mathcal{H}$ is an RKHS with kernel $k$ and canonical feature map $\Phi(x) = k(x, \cdot)$,[2] and (iii) all kernels are assumed to be bounded.[3] Our main results (Theorem 2 and Theorem 3) are that in this case the expected kernel trick applies to both items in the kernelized version of Theorem 1. The key to these results is an expression for inner products of cumulants in RKHSs (Lemma 1).

**A combinatorial expression of cumulants.**   Classical cumulants can be defined via the moment generating function or via combinatorial sums over *partitions* (Appendix C.1). To generalize cumulants to RKHSs the combinatorial definition is the most efficient way. A partition $\pi$ of $m$ elements is a family of non-empty, disjoint subsets $\pi_1, \ldots, \pi_b$ of $\{1, \ldots, m\}$ whose union is the whole set; formally $\bigcup_{j=1}^b \pi_j = \{1, \ldots, m\}$ and $\pi_i \cap \pi_j = \varnothing$ for $i \neq j$. We call $b$ the number of blocks of the partition $\pi$ and use the shorthand $|\pi|$ to denote it. The set of all partitions of $m$ is denoted with $P(m)$. To formulate our main results, it is convenient to associate with a measure $\gamma$ and a partition $\pi$ the so-called partition measure $\gamma_\pi$ that is given by permuting the marginals of $\gamma$.

**Definition 2** (Partition measure). *Let $\gamma$ be a probability measure on $\mathcal{X}_1 \times \cdots \times \mathcal{X}_d$ and $\pi \in P(d)$. Define*

$$\gamma_\pi := \gamma|_{\mathcal{X}_{\pi_1}} \otimes \cdots \otimes \gamma|_{\mathcal{X}_{\pi_b}},$$

*where $\mathcal{X}_{\pi_i}$ denotes the product space $\prod_{j \in \pi_i} \mathcal{X}_j$ and $\gamma|_{\mathcal{X}_{\pi_i}}$ is the corresponding marginal distribution of $\gamma$. We call $\gamma_\pi$ the partition measure induced by $\pi$.*

We also associate with $\gamma$ and a multi-index $\mathbf{i}$ the so-called diagonal measure $\gamma^{\mathbf{i}}$ that is given by repeating marginals according to $\mathbf{i}$.

**Definition 3** (Diagonal measure). *Let $\gamma$ be a probability measure on $\mathcal{X}_1 \times \cdots \times \mathcal{X}_d$ and $\mathbf{i} = (i_1, \ldots, i_d) \in \mathbb{N}^d$. Define*

$$\gamma^{\mathbf{i}} := \mathrm{Law}(\underbrace{X_1, \ldots, X_1}_{i_1 \text{ times}}, \underbrace{X_2, \ldots, X_2}_{i_2 \text{ times}}, \ldots, \underbrace{X_d, \ldots, X_d}_{i_d \text{ times}}),$$

*where $(X_1, \ldots, X_d) \sim \gamma$. We call $\gamma^{\mathbf{i}}$ the diagonal measure induced by $\mathbf{i}$.*

In general, the partition measure $\gamma_\pi$ and the diagonal measure are not probability measures on $\mathcal{X}_1 \times \cdots \times \mathcal{X}_d$ but on spaces that are constructed by permuting or repeating $\mathcal{X}_1, \ldots, \mathcal{X}_d$. Formally, $\gamma_\pi$ is a probability measure on $\mathcal{X}_{\pi_1} \times \cdots \times \mathcal{X}_{\pi_b}$ and $\gamma^{\mathbf{i}}$ is a probability measure on $\mathcal{X}_1^{i_1} \times \cdots \times \mathcal{X}_d^{i_d}$; thus, $\gamma_\pi$ has $d$ coordinates and $\gamma^{\mathbf{i}}$ has $\deg(\mathbf{i})$ coordinates. These two constructions can be combined, writing $\gamma_\pi^{\mathbf{i}}$ for the measure $(\gamma^{\mathbf{i}})_\pi$ which makes sense whenever $\pi \in P(\deg(\mathbf{i}))$. We can now write down our generalization of cumulants.

---

[2]$k(x, \cdot)$ denotes the function $x' \mapsto k(x, x')$ with $x \in \mathcal{X}$ fixed.

[3]A kernel $k : \mathcal{X} \times \mathcal{X} \to \mathbb{R}$ is called bounded if there exists $B \in \mathbb{R}$ such that $\sup_{x, x' \in \mathcal{X}} k(x, x') \leq B$.

**Definition 4** (Kernelized cumulants). *Let $\gamma$ be a probability measure on $\mathcal{X}_1 \times \cdots \times \mathcal{X}_d$ and let $(\mathcal{H}_1, k_1), \ldots, (\mathcal{H}_d, k_d)$ be RKHSs on $\mathcal{X}_1, \ldots, \mathcal{X}_d$ respectively. We define the kernelized cumulants*

$$\kappa_{k_1,\ldots,k_d}(\gamma) := \left(\kappa_{k_1,\ldots,k_d}^{\mathbf{i}}(\gamma)\right)_{\mathbf{i} \in \mathbb{N}^d} \in \mathrm{T}$$

*as follows*

$$\kappa_{k_1,\ldots,k_d}^{\mathbf{i}}(\gamma) := \sum_{\pi \in P(m)} c_\pi \mathbb{E}_{\gamma_\pi^{\mathbf{i}}} k^{\otimes \mathbf{i}}((X_1, \ldots, X_m), \cdot),$$

*where $m = \deg(\mathbf{i})$, $c_\pi := (-1)^{|\pi|-1}(|\pi|-1)!$, $\gamma_\pi^{\mathbf{i}} = (\gamma^{\mathbf{i}})_\pi$ and*

$$k^{\otimes \mathbf{i}}((x_1, \ldots, x_m), (y_1, \ldots, y_m)) := k_1(x_1, y_1) \cdots k_1(x_{i_1}, y_{i_1}) \tag{4}$$
$$\cdots k_d(x_{m-i_d+1}, y_{m-i_d+1}) \cdots k_d(x_m, y_m)$$

*is the reproducing kernel of $\mathcal{H}^{\otimes \mathbf{i}}$ where $\mathcal{H} = \mathcal{H}_1 \times \cdots \times \mathcal{H}_d$.*

Def. 4 is the natural generalization of the combinatorial definition of cumulants in $\mathbb{R}^d$ and Appendix C.2 gives an equivalent definition via a generating function analogous to (2). However, our posthoc justification that these are the "right" definitions for cumulants in an RKHS are Theorems 2 and 3 that show that these kernelized cumulants have the same powerful properties as classic cumulants in $\mathbb{R}^d$ (Theorem 1).

**Example 3.1** (Kernelized cumulants). *Let $\gamma$ be a probability measure on $\mathcal{X}_1 \times \mathcal{X}_2$, with the RKHSs $(\mathcal{H}_1, k_1), (\mathcal{H}_2, k_2)$ given. Denote the random variables $K_1 = k_1(X_1, \cdot), K_2 = k_2(X_2, \cdot)$ where $(X_1, X_2) \sim \gamma$. Then the degree two kernelized cumulants are given as $\kappa_{k_1,k_2}^{(2,0)}(\gamma) = \mathbb{E}\left[K_1^{\otimes 2}\right] - \mathbb{E}\left[K_1\right]^{\otimes 2}, \kappa_{k_1,k_2}^{(1,1)}(\gamma) = \mathbb{E}\left[K_1 \otimes K_2\right] - \mathbb{E}\left[K_1\right] \otimes \mathbb{E}\left[K_2\right], \kappa_{k_1,k_2}^{(0,2)}(\gamma) = \mathbb{E}\left[K_2^{\otimes 2}\right] - \mathbb{E}\left[K_2\right]^{\otimes 2}$.*

**Inner products of cumulants.** Computing inner products of moments is straightforward thanks to a nonlinear kernel trick, see Lemma 6 in the Appendix. For example, given two probability measures $\gamma_1, \gamma_2$ with corresponding random variables $(X_1, \ldots, X_d) \sim \gamma_1, (Y_1, \ldots, Y_d) \sim \gamma_2$ on $\mathcal{X}_1 \times \cdots \times \mathcal{X}_d$ and RKHSs $(\mathcal{H}_1, k_1), \ldots, (\mathcal{H}_d, k_d)$ on $\mathcal{X}_1, \ldots, \mathcal{X}_d$ with bounded kernels, and $\mathcal{H} = \mathcal{H}_1 \times \cdots \times \mathcal{H}_d$, we can express:

$$\langle \mu_{k_1,\ldots,k_d}^{\mathbf{i}}(\gamma_1), \mu_{k_1,\ldots,k_d}^{\mathbf{i}}(\gamma_2) \rangle_{\mathcal{H}^{\otimes \mathbf{i}}} = \mathbb{E}_{\gamma_1 \otimes \gamma_2} k_1(X_1, Y_1)^{i_1} \cdots k_d(X_d, Y_d)^{i_d}, \tag{5}$$

where $\mu_{k_1,\cdots,k_d}^{\mathbf{i}}$ is defined in Def. 1, and the expectation is taken over the product measure $\gamma_1 \otimes \gamma_2$.

**Example 3.2.** *In the particular case of $d = 1$, (5) reduces to the well-known formula for the inner product of mean embeddings $\langle \mu_k^{(1)}(\gamma_1), \mu_k^{(1)}(\gamma_2) \rangle_{\mathcal{H}_k} = \mathbb{E}_{\gamma_1 \otimes \gamma_2} k(X, Y)$.*

**Lemma 1** (Inner product of cumulants). *Let $(\mathcal{H}_1, k_1), \ldots, (\mathcal{H}_d, k_d)$ be RKHSs with bounded kernels on $\mathcal{X}_1, \ldots, \mathcal{X}_d$ respectively, and let $\gamma$ and $\eta$ two probability measures on $\mathcal{X}_1 \times \cdots \times \mathcal{X}_d$, $\mathbf{i} = (i_1, \ldots, i_d) \in \mathbb{N}^d$ such that $\deg(\mathbf{i}) = m$. Then*

$$\langle \kappa_{k_1,\ldots,k_d}^{\mathbf{i}}(\gamma), \kappa_{k_1,\ldots,k_d}^{\mathbf{i}}(\eta) \rangle_{\mathcal{H}^{\otimes \mathbf{i}}} = \sum_{\pi, \tau \in P(m)} c_\pi c_\tau \mathbb{E}_{\gamma_\pi^{\mathbf{i}} \otimes \eta_\tau^{\mathbf{i}}} k^{\otimes \mathbf{i}}((X_1, \ldots, X_m), (Y_1, \ldots, Y_m)).$$

**Point separating kernels.** In the classic MMD setting the injectivity of the mean embedding $\gamma \mapsto \mathbb{E}_{X \sim \gamma}[k(X, \cdot)]$ on probability measures (known as the characteristic property of the kernel $k$) is equivalent to the MMD being a metric; this property is central in applications. We formulate our theoretical results in the next section using the much weaker property of what we term "point-separating" which is satisfied for essentially all popular kernels.

**Definition 5** (Point-separating kernel). *We call a kernel $k : \mathcal{X} \times \mathcal{X} \to \mathbb{R}$ point-separating if the canonical feature map $\Phi : x \mapsto k(x, \cdot)$ is injective.*

### 3.1 (Semi-)metrics for probability measures

In this section we use cumulants to characterize probability measures and show how to compute the distance between kernelized cumulants with the expected kernel trick.

**Theorem 2** (Characterization of distributions with cumulants). *Let $\gamma$ and $\eta$ be two probability measures on $\mathcal{X}_1 \times \cdots \times \mathcal{X}_d$, $(\mathcal{H}_1, k_1), \ldots, (\mathcal{H}_d, k_d)$ RKHSs on the Polish spaces $\mathcal{X}_1, \ldots, \mathcal{X}_d$ such that for every $1 \le j \le d$ $k_j$ is a bounded, continuous, point-separating kernel. Then*

$$\gamma = \eta \text{ if and only if } \kappa_{k_1,\ldots,k_d}(\gamma) = \kappa_{k_1,\ldots,k_d}(\eta).$$

*Moreover, the expected kernel trick applies and for $\mathbf{i} \in \mathbb{N}^d$ with $\deg(\mathbf{i}) = m$, and $k^{\otimes \mathbf{i}}$ and $\mathcal{H}^{\otimes \mathbf{i}}$ as in* (4)

$$
\begin{aligned}
d^{\mathbf{i}}(\gamma, \eta) &:= \|\kappa_{k_1,\ldots,k_d}^{\mathbf{i}}(\gamma) - \kappa_{k_1,\ldots,k_d}^{\mathbf{i}}(\eta)\|_{\mathcal{H}^{\otimes \mathbf{i}}}^2 \qquad\qquad (6) \\
&= \sum_{\pi,\tau \in P(m)} c_\pi c_\tau \Big[ \mathbb{E}_{\gamma_\pi^{\mathbf{i}} \otimes \gamma_\tau^{\mathbf{i}}} k^{\otimes \mathbf{i}}((X_1, \ldots, X_m), (Y_1, \ldots, Y_m)) \\
&\qquad\qquad\qquad + \mathbb{E}_{\eta_\pi^{\mathbf{i}} \otimes \eta_\tau^{\mathbf{i}}} k^{\otimes \mathbf{i}}((X_1, \ldots, X_m), (Y_1, \ldots, Y_m)) \\
&\qquad\qquad\qquad - 2\mathbb{E}_{\gamma_\pi^{\mathbf{i}} \otimes \eta_\tau^{\mathbf{i}}} k^{\otimes \mathbf{i}}((X_1, \ldots, X_m), (Y_1, \ldots, Y_m)) \Big].
\end{aligned}
$$

We recall Example 3.1 and now give examples of distances between such expressions

**Example 3.3** ($m = 1$). *Applied with $m = 1$ and $d = 1$,* (6) *becomes* $\mathrm{MMD}_k^2(\gamma, \eta)$

$$\|\kappa_k^{(1)}(\gamma) - \kappa_k^{(1)}(\eta)\|_{\mathcal{H}_k}^2 = \mathbb{E}k(X, X') + \mathbb{E}k(Y, Y') - 2\mathbb{E}k(X, Y),$$

*where $X, X'$ denotes independent copies of $\gamma$ and $Y, Y'$ denotes independent copies of $\eta$.*

**Example 3.4** ($m = 2$). *For $m = 2$ and $d = 1$,* (6) *reduces to*

$$
\begin{aligned}
\|\kappa_k^{(2)}(\gamma) - \kappa_k^{(2)}(\eta)\|_{\mathcal{H}^{(1,1)}}^2 &= \mathbb{E}k(X, X')k(X'', X''') + \mathbb{E}k(Y, Y')k(Y'', Y''') + \mathbb{E}k(X, X')^2 \\
&+ \mathbb{E}k(Y, Y')^2 + 2\mathbb{E}k(X, Y)k(X', Y) + 2\mathbb{E}k(X, Y)k(X, Y') - 2\mathbb{E}k(X, Y)k(X', Y') \\
&- 2\mathbb{E}k(X, Y)^2 - 2\mathbb{E}k(X, X')k(X, X'') - 2\mathbb{E}k(Y, Y')k(Y, Y''),
\end{aligned}
$$

*where $X, X', X'', X'''$ denotes independent copies of $\gamma$ and $Y, Y', Y'', Y'''$ denotes independent copies of $\eta$. This expression compares the variances in the RKHS instead of the means. This is an example of the* kernel variance embedding *defined in the next subsection.*

The price for the weak assumption of a point-separating kernel is that without any stronger assumptions one does not get a metric in general, and the all-purpose way to achieve a metric is to take an infinite sum over all $d^{\mathbf{i}}$'s. If we only use the degree $m = 1$ term $d^{\mathbf{i}}$ reduces to the well-known MMD formula which requires characteristicness to become a metric (see Example 3.3). There are two reasons why working under weaker assumptions is useful: firstly, if the underlying kernel is not characteristic this sum gives a structured way to incorporate finer information that discriminates the two distributions; an extreme case is the linear kernel $k(x, y) = \langle x, y \rangle$ which is point-separating, and in this case the sum reduces to the differences of classical cumulants. Secondly, under the stronger assumption of characteristicness one already has a metric after truncation at degree $m = 1$ (the classical MMD). However, in the finite-sample case adding higher degree terms can lead to increased power. Indeed, our experiments (Section 4) show that even just going one degree further (i.e. taking $m = 2$), can lead to more powerful tests.

## 3.2 A characterization of independence

Here we characterize independence in terms of kernelized cumulants.

**Theorem 3** (Characterization of independence with cumulants). *Let $\gamma$ be a probability measure on $\mathcal{X}_1 \times \cdots \times \mathcal{X}_d$, and $(\mathcal{H}_1, k_1), \ldots, (\mathcal{H}_d, k_d)$ RKHSs on Polish spaces $\mathcal{X}_1, \ldots, \mathcal{X}_d$ such that for every $1 \le j \le d$ $k_j$ is a bounded, continuous, point-separating kernel. Then*

$$\gamma = \gamma|_{\mathcal{X}_1} \otimes \cdots \otimes \gamma|_{\mathcal{X}_d} \text{ if and only if } \kappa_{k_1,\ldots,k_d}^{\mathbf{i}}(\gamma) = 0$$

*for every $\mathbf{i} \in \mathbb{N}_+^d$. Moreover, the expected kernel trick applies in the sense that for $\mathbf{i} \in \mathbb{N}_+^d$*

$$\|\kappa_{k_1,\ldots,k_d}^{\mathbf{i}}(\gamma)\|_{\mathcal{H}^{\otimes \mathbf{i}}}^2 = \sum_{\pi,\tau \in P(m)} c_\pi c_\tau \mathbb{E}_{\gamma_\pi^{\mathbf{i}} \otimes \gamma_\tau^{\mathbf{i}}} k^{\otimes \mathbf{i}}((X_1, \ldots, X_m), (Y_1, \ldots, Y_m)), \qquad (7)$$

*where $m := \deg(\mathbf{i})$, and $k^{\otimes \mathbf{i}}$ and $\mathcal{H}^{\otimes \mathbf{i}}$ are defined as in* (4).

Applied to $\mathbf{i} = (1,1)$, the expression (7) reduces to the classical HSIC for two components, see Example 3.5 below. But for general $\mathbf{i}$ this construction leads to genuine new statistics in RKHSs.

**Example 3.5** (Specific case: HSIC, kernel Lancaster interaction, kernel Streitberg interaction)**.** *If $d = 2$ there is only one order 2 index in $\mathbb{N}_+^d$, namely $\mathbf{i} = (1,1)$; in this case (7) reduces to the classical HSIC equation*

$$\|\kappa_{k_1,k_2}^{(1,1)}(\gamma)\|_{\mathcal{H}^{(1,1)}}^2 = \mathbb{E}k_1(X,Y)k_2(X,Y) + \mathbb{E}k_1(X,Y)k_2(X',Y') - 2\mathbb{E}k_1(X,Y)k_2(X',Y),$$

*where $(X,Y)$ and $(X',Y')$ are independent copies of the same random variable following $\gamma$. More generally, with $\mathbf{i} = \mathbf{1}_d$ one gets the kernel Streitberg interaction (Streitberg, 1990; Sejdinovic et al., 2013a; Liu et al., 2023), and specifically the kernel Lancaster interaction (Sejdinovic et al., 2013a) for $d \in \{2,3\}$; the latter reduces to HSIC for two random variables ($d = 2$).*

### 3.3 Finite-sample statistics

To apply Theorem 2 and Theorem 3 in practice, one needs to estimate expressions such as $\mathbb{E}k^{\otimes\mathbf{i}}((X_1,\ldots,X_m),(Y_1,\ldots,Y_m))$. One could use classical estimators such as *U-statistic* (Van der Waart, 2000) which lead to unbiased estimators. However, we follow Gretton et al. (2008) and use a *V-statistic* which is biased but conceptually simpler, easier, and efficient to compute. We note that the estimators presented here all have quadratic complexity like MMD and HSIC, see Appendix E.

**A two-sample test for non-characteristic feature maps.** If $k$ is characteristic then $\mathrm{MMD}_k(\gamma,\eta) = 0$ exactly when $\gamma = \eta$, but we can still increase testing power by considering the distance between the kernel variance and skewness embeddings, which leads us to use our semi-metrics $d^{(2)}(\gamma,\eta)$ and $d^{(3)}(\gamma,\eta)$ as defined in (6). An efficient estimator for $d^{(3)}$ is given in detail in Appendix E; we provide the full expression for $d^{(2)}$ here.

**Lemma 2** ($d^{(2)}$ estimation, see (6))**.** *The V-statistic for $d^{(2)}(\gamma,\eta) = \|\kappa_k^{(2)}(\gamma) - \kappa_k^{(2)}(\eta)\|_{\mathcal{H}^{(1,1)}}^2$ is*

$$\frac{1}{N^2}\mathrm{Tr}\big[(\mathbf{K}_x\mathbf{J}_N)^2\big] + \frac{1}{M^2}\mathrm{Tr}\big[(\mathbf{K}_y\mathbf{J}_M)^2\big] - \frac{2}{NM}\mathrm{Tr}\big[\mathbf{K}_{xy}\mathbf{J}_M\mathbf{K}_{xy}^\top\mathbf{J}_N\big],$$

*where $\mathrm{Tr}$ denotes trace, $(x_n)_{n=1}^N \overset{\mathrm{i.i.d.}}{\sim} \gamma$, $(y_m)_{m=1}^M \overset{\mathrm{i.i.d.}}{\sim} \eta$, $\mathbf{K}_x = [k(x_i,x_j)]_{i,j=1}^N \in \mathbb{R}^{N\times N}$, $\mathbf{K}_y = [k(y_i,y_j)]_{i,j=1}^M \in \mathbb{R}^{M\times M}$, $\mathbf{K}_{x,y} = [k(x_i,y_j)]_{i,j=1}^{N,M} \in \mathbb{R}^{N\times M}$, $\mathbf{J}_n = \mathbf{I}_n - \frac{1}{n}\mathbf{1}_n\mathbf{1}_n^\top \in \mathbb{R}^{n\times n}$, with $\mathbf{1}_n = (1,\ldots,1) \in \mathbb{R}^n$.*

**A kernel independence test.** By Theorem 3, if $\gamma = \gamma|_{\mathcal{X}_1} \otimes \gamma|_{\mathcal{X}_2}$, then $\kappa^{(2,1)}(\gamma) = 0$ and $\kappa^{(1,2)}(\gamma) = 0$. We may compute the magnitude of either $\kappa^{(2,1)}(\gamma)$ or $\kappa^{(1,2)}(\gamma)$ – we will refer to these quantities as *cross skewness independence criterion* (CSIC). Note that these criteria are asymmetric. When $d = 2$ we have a probability measure $\gamma$ on $\mathcal{X}_1 \times \mathcal{X}_2$ and two kernels $k : \mathcal{X}_1^2 \to \mathbb{R}$, $\ell : \mathcal{X}_2^2 \to \mathbb{R}$. Assume that we have samples $(x_i,y_i)_{i=1}^N$ and use the shorthand notation $\mathbf{K} = \mathbf{K}_x, \mathbf{L} = \mathbf{L}_y$ (similarly to Lemma 2) and $\mathbf{H} = \mathbf{H}_N = \frac{1}{N}\mathbf{1}_N\mathbf{1}_N^\top \in \mathbb{R}^{N\times N}$. Denote by $\circ$ the Hadamard product and $\langle\cdot\rangle$ the sum over all elements of a matrix. Then one can derive the following CSIC estimator.(Note that matrix multiplication takes precedence over the Hadamard product.)

**Lemma 3** (CSIC estimation)**.** *The V-statistic for $\|\kappa_{k,\ell}^{(1,2)}(\gamma)\|_{\mathcal{H}_k^{\otimes 1}\otimes\mathcal{H}_\ell^{\otimes 2}}^2$ is*

$$\frac{1}{N^2}\bigg\langle \mathbf{K}\circ\mathbf{K}\circ\mathbf{L} - 4\mathbf{K}\circ\mathbf{K}\mathbf{H}\circ\mathbf{L} - 2\mathbf{K}\circ\mathbf{K}\circ\mathbf{L}\mathbf{H} + 4\mathbf{K}\mathbf{H}\circ\mathbf{K}\circ\mathbf{L}\mathbf{H}$$

$$+ 2\mathbf{K}\circ\mathbf{L}\left\langle\frac{\mathbf{K}}{N^2}\right\rangle + 2\mathbf{K}\mathbf{H}\circ\mathbf{H}\mathbf{K}\circ\mathbf{L} + 4\mathbf{K}\circ\mathbf{H}\mathbf{K}\circ\mathbf{L}\mathbf{H} + \mathbf{K}\circ\mathbf{K}\left\langle\frac{\mathbf{L}}{N^2}\right\rangle$$

$$- 8\mathbf{K}\circ\mathbf{L}\mathbf{H}\left\langle\frac{\mathbf{K}}{N^2}\right\rangle - 4\mathbf{K}\circ\mathbf{H}\mathbf{K}\left\langle\frac{\mathbf{L}}{N^2}\right\rangle + 4\left\langle\frac{\mathbf{K}}{N^2}\right\rangle^2\mathbf{L}\bigg\rangle.$$

**Remark (computational complexity w.r.t. degree $m$).** We saw that the computational complexity of the cumulant based measures is quadratic w.r.t. the sample size. Let $B_m = |P(m)|$ be the $m$-th

Bell number, in other words the number of elements in $P(m)$. The Bell numbers follow a recursion: $B_{m+1} = |P(m+1)| = \sum_{k=0}^{m} \binom{m}{k} B_k$, with the first elements of the sequence being $B_0 = B_1 = 1$, $B_2 = 2$, $B_3 = 5$, $B_4 = 15$, $B_5 = 52$. By (6)-(7), in the worst case the number of operations to compute $d^{\mathbf{i}}(\gamma, \eta)$ or $\|\kappa_{k_1,\ldots,k_d}^{\mathbf{i}}(\gamma)\|_{\mathcal{H}^{\otimes i}}^2$ ($m = \deg(\mathbf{i})$) is proportional to $B_m^2$ (it equals to $3B_m^2$ and to $B_m^2$, respectively). Though asymptotically $B_m$ grows quickly (de Bruijn, 1981; Lovász, 1993), for reasonably small degrees the computation is still manageable. In addition, merging various terms in the estimator can often be carried out, which leads to computational saving. For instance, the estimator of $d^{(2)}$ (see Lemma 2, Example E.1), CSIC (Lemma 3, Example E.2) and $d^{(3)}$ (Example E.3) consists of only 3, 11 and $10 + 2 \times 7 = 24$ terms compared to the predicted worst-case setting of $3B_2^2 = 12$, $B_3^2 = 25$, and $3B_3^2 = 75$ terms, respectively. On a practical side, we found that using $m \in \{2, 3\}$ is a good compromise between gain in sample efficiency and ease of implementation.

## 4  Experiments

In this section, we demonstrate the efficiency of the proposed kernel cumulants in two-sample and independence testing.[4]

- Two-sample test: Given $N - N$ samples from two probability measures $\gamma$ and $\eta$ on a space $\mathcal{X}$, the goal was to test the null hypothesis $H_0 : \gamma = \eta$ against the alternative $H_1 : \gamma \neq \eta$. The compared test statistics ($S$) were MMD, $d^{(2)}$, and $d^{(3)}$.

- Independence test: Given $N$ paired samples from a probability measure $\gamma$ on a product space $\mathcal{X}_1 \times \mathcal{X}_2$, the aim was to the test the null hypothesis $H_0 : \gamma = \gamma_1 \otimes \gamma_2$ against the alternative $H_1 : \gamma \neq \gamma_1 \otimes \gamma_2$. The compared test statistics ($S$) were HSIC and CSIC.

In our experiments $H_1$ held, and the estimated power of the tests is reported. Permutation test was applied to approximate the null distribution and its 0.95-quantile (which corresponds to the level choice $\alpha = 0.05$): We first computed our test statistic $S$ using the given samples ($S_0 = S$), and then permuted the samples 100 times. If $S_0$ was in a high percentile ($\geq 95\%$ in our case) of the resulting distribution of $S$ under the permutations, we rejected the null. We repeated these experiments 100 times to estimate the power of the test. This procedure was in turn repeated 5 times and the 5 samples are plotted as a box plot along with a line plot showing the mean against the number of samples ($N$) used. All experiments were performed using the rbf-kernel $\text{rbf}_\sigma(\mathbf{x}, \mathbf{y}) = e^{-\frac{\|\mathbf{x}-\mathbf{y}\|_2^2}{2\sigma^2}}$, where the parameter $\sigma$ is called the *bandwidth*. We performed all experiments for every bandwidth of the form $\sigma = a10^b$ where $a = 1, 2.5, 5, 7.5$ and $b = -5, -4, -3, -2, -1, 0$ and the optimal value across the bandwidths was chosen for each method and sample size. The experiments were carried out on a laptop with an i7 CPU and 16GBs of RAM.

### 4.1  Synthetic data

For synthetic data we designed two experiments.

- 2-sample test: We compared a uniform distribution with a mixture of two uniforms.

- Independence test: We considered the joint measure of a uniform and a correlated $\chi^2$ random variable. We also use this same benchmark to compare the efficiency of classical and kernelized cumulants in Appendix D.

**Comparing a uniform with a mixture of uniforms.**  Even for simpler distributions like mixtures of uniform distributions it can be hard to pick up higher-order features, and $d^{(2)}$ can outperform MMD even when provided with a moderate number of samples. Here we compared one uniform distribution $U[-1, 1]$ with an equal mixture of $U[0.35, 0.778]$ and $U[-0.35, -0.778]$. The endpoints in the mixture were chosen to match the first three moments of $U[-1, 1]$. The number of samples used ranged from 5 to 50, and the results are summarized in Fig. 1. One can see that with $d^{(2)}$ the power approaches $100\%$ much faster than with using MMD.

---

[4]All the code replicating our experiments is available at `https://github.com/PatricBonnier/Kernelized-Cumulants`.

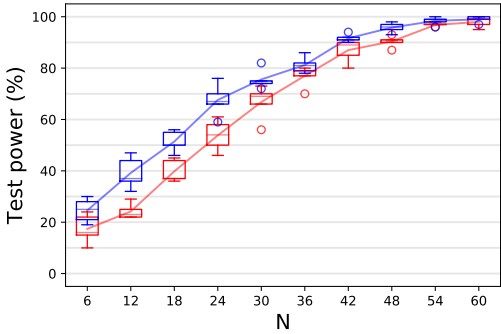

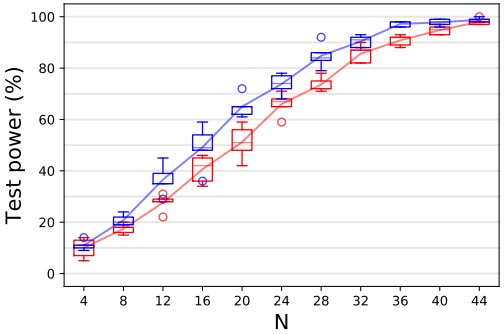

Figure 2: Test power as a function of the sample size ($N$) of independence testing using HSIC (red) and CSIC statistics (blue), with the independence testing between $Y_{0.5}^2$ and $X$.

Figure 3: Two-sample testing using MMD (red) and $d^{(2)}$ (blue) on the Seoul bicycle data set.

**Independence between a uniform and a $\chi^2$.**
Let $X \sim U[0,1]$ and $Z \sim N(0,1)$ be independent of $X$. Denote by $\Phi$ the c.d.f. of a standard normal distribution and define $Y_p$ to be a mixture with weight $p$ at $\Phi^{-1}(X)$ and weight $1 - p$ at $Z$ so that $Y_0 = Z$ and $Y_1 = \Phi^{-1}(X)$. We test for independence for $p = 0.5$ between $Y_p^2$—which will be $\chi^2$ distributed with 1 degree of freedom–and $X$. As the statistical dependence of $Y_p^2$ and $X$ is more complicated than a simple correlation we expect that higher-order features of the data will help in the independence testing. The number of samples used ranged from 6 to 60, with the results summarized in Fig. 2. One can see that CSIC supersedes HSIC for every sample size, and the difference is more pronounced for smaller ones.

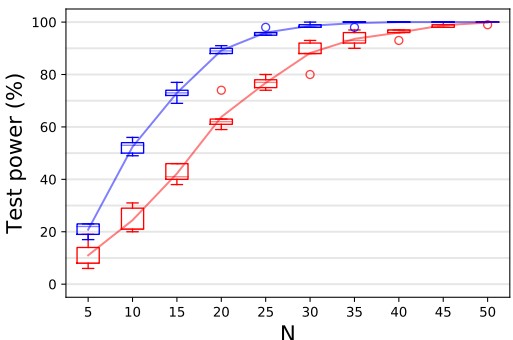

Figure 1: Test power as a function of the sample size ($N$) of two-sample test using the MMD (red) and $d^{(2)}$ statistics (blue), with $U[-1,1]$ and an equal mixture of $U[0.35, 0.778]$ and $U[-0.35, -0.778]$ compared.

## 4.2 Real-world data

We demonstrate the efficiency of the kernelized cumulants on real-world data. We designed two experiments.

- 2-sample test: Here the goal was to test if environmental data in two different seasons, and traffic data at different speeds can be distinguished.

- independence test: The aim was to test if two distributions describing traffic flow and other traffic factors are independent.

To improve performance of both test statistics, all features are standardized to lie between $0$ and $1$.

**Seoul bicycle data.** The Seoul bicycle data set (E et al., 2020) consists of environmental data along with the number of bicycle rentals. The environmental data consists of 9 numerical values, 1 categorical value (season), and two binary values. We compare the distribution of the environmental data in the winter and the fall, as we expect these distributions to be different. Concretely, we do 2-sample testing on two measures $\gamma, \eta$ on $\mathbb{R}^{11}$, and assume that $\gamma \neq \eta$ where $\gamma$ is the distribution of the environmental data in winter, and $\eta$ is that of the data in the fall. Permutation testing was performed for $N$ between 4 and 44, with results summarized in Fig. 3. As it can be observed that $d^{(2)}$ outperforms MMD in terms of test power. For the Type I error, i.e. the probability of falsely rejecting the null hypothesis (comparing winter data with itself), it hovers between $5 - 10\%$ for both statistics,

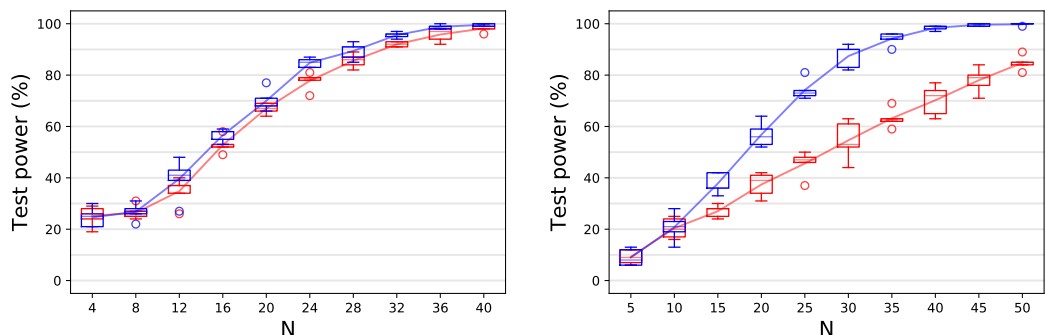

Figure 4: Independence testing using HSIC (red) and CSIC (blue) on the Sao Paulo traffic dataset.

Figure 5: Two-sample testing using MMD (red) and $d^{(3)}$ (blue) on the Sao Paulo traffic dataset.

which is admittedly slightly higher than the desired $5\%$ due to the small sample size, but very similar for both statistics; for further details, the reader is referred to Fig. 8 in Appendix D.

**Brazilian traffic data.** We used the Sao Paulo traffic benchmark (Ferreira, 2016) to perform independence testing. The dataset consists of 16 different integer-valued statistics about the hourly traffic in Sao Paulo such as blockages, fires and other reasons that might hold up traffic. This is combined with a number that describes the slowness of traffic at the given hour; so $\mathcal{X}_1 = \mathbb{R}^{16}$, $\mathcal{X}_2 = \mathbb{R}$. One expects a strong dependence between the two sets—or equivalently, for the null hypothesis to be false—and for the statistics are heavily skewed towards 0 as it is naturally sparse. For independence testing we performed permutation testing for $N$ between 4 and 40. The resulting test powers are summarized in Fig. 4. As it can be seen, HSIC and CSIC performs similarly for very low sample sizes, but for anything else CSIC is the favorable statistic in terms of test power. For two-sample testing, we sampled $N$ between 5 and 50 and compared the distribution of slow moving traffic with the fast moving traffic. The results are summarized in Fig. 5. It is clear that $d^{(3)}$ performs similarly to MMD in terms of test power for very small sample sizes, but significantly better for larger ones.

## 5 Conclusion

We defined cumulants for random variables in RKHSs by extending the algebraic characterization of cumulants on $\mathbb{R}^d$. This construction results in a structured description of the law of random variables that goes beyond the classic kernelized mean and covariance. A kernel trick allows us to compute the resulting kernelized cumulants. We applied our theoretical results to two-sample and independence testing; although kernelized mean and covariance are sufficient for this task, the higher-order kernelized cumulants have the potential to increase the test power and to relax the assumptions on the kernel. Our experiments on real and synthetic data show that kernelized cumulants can indeed lead to significant improvement of the test power. A disadvantage of these higher-order statistics is that their theoretical analysis requires more mathematical machinery although we emphasize that the resulting estimators are simple V-statistics.

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

## A  Moments and cumulants

Already for real-valued random variable $X$, moments have well-known drawbacks that make cumulants often preferable as statistics. For a detailed introduction to the use of cumulants in statistics we refer to McCullagh (2018). Here we just mention that

1. the moment generating function $f(t) = \mathbb{E}[e^{tX}] = \sum_m \mu_m t^m / m!$ describes the law of $X$ with sequence $(\mu_m)$ of moments $\mu_m = \mathbb{E}[X^m] \in \mathbb{R}$. However, since the function $t \mapsto f(t)$ is the expectation of an exponential, one would often expect that $f$ is also "exponential in $t$", hence $g(t) = \log f(t) = \sum_m \kappa_m \frac{t^m}{m!}$ should be simpler to describe as a power series. For example, for a Gaussian $f(t) = e^{t\mathbb{E}(X) + \frac{t^2}{2} Var(X)}$ and while $\mu_m$ can be in this case explicitly calculated and uneven moments vanish, the $m$-moments are fairly complicated compared to the power series expansion of $g(t) = \kappa_1 t + \kappa_2 \frac{t^2}{2}$ which just consists of $\kappa_1$ (mean) and $\kappa_2$ (variance).

2. In the moment sequence $\mu_m$, lower moments can dominate higher moments. Hence, a natural idea to compensate for these "different scales" is to systematically subtract lower moments from higher moments. As mentioned in the introduction, this is in particular troublesome if finite samples are available. Even in dimension $d = 1$ the second moment is dominated by the squared mean, that is for a real-valued random variable $X \sim \gamma$

$$\mu^2(\gamma) = (\mu^1(\gamma))^2 + \mathrm{Var}(X),$$

where $\mathrm{Var}(X) := \mathbb{E}[(X - \mu^1(\gamma))^2]$. It is well known that the minimum variance unbiased estimators for the variance are more efficient than that for the second moment: denoting them by $\widehat{\mu^2}$ and $\widehat{\kappa}$ respectively, one can show (Bonnier and Oberhauser, 2020) that given $N$ samples from $X$, the following holds

$$\mathrm{Var}\left(\widehat{\mu^2}\right) = \mathrm{Var}\left(\widehat{\kappa}\right) + \frac{2}{N}\left[(\mathbb{E}X)^4 - (\mathbb{E}X)^2 \mathrm{Var}(X) - 2\frac{\mathrm{Var}(X)^2}{N-1}\right].$$

This means that when $X$ has a large mean, it is more efficient to estimate its variance than its second moment since the last term in the above expression dominates. Hence, the variance $\mathrm{Var}(X)$ is typically a much more sensible second-order statistic than $\mu^2(\gamma)$. However, we emphasize that there are many other reasons why cumulants can have better properties as estimators

3. Cumulants characterize laws and the independence of two random variables manifests itself simply as vanishing of cross-cumulants. In view of the above item 2, this means for example that testing independence can be preferable in terms of vanishing cumulants rather than testing if moments factor $\mathbb{E}[X^m Y^n] = \mathbb{E}[X^m]\mathbb{E}[X^n]$, and similarly for testing if distributions are the same.

The caveat to the above points is that it is not true that cumulants are always preferable. For example, there are distributions for which (a) the moment generating function is not naturally exponential in $t$, (b) lower moments do not dominate higher moments, (c) consequently independence or two-sample testing become worse with cumulants. While one can write down conditions under which for example, the variance of the kernelized cumulants is lower, the use of cumulants among statisticians is to simply regard cumulants as arising from natural motivations which leads to another estimator in their toolbox.

The main idea of our paper is simply that for the same reasons that cumulants can turn out to be powerful for real or vector-valued random variables, cumulants of RKHS-valued random variables are a natural choice of statistics. The situation is more complicated since it requires formalizing moment- and cumulant-generating functions in RKHS but ultimately a kernel trick allows for circumventing the computational bottleneck of working in infinite dimensions and leads to computable estimators for independence and two-sample testing.

Further, we note that although cumulants are classic for vector-valued data, there seems to be not much work done about extending their properties to general structured data. Our kernelized cumulants apply to any set $\mathcal{X}$ where a kernel is given. This includes many practically relevant examples such as strings (Lodhi et al., 2002), graphs (Kriege et al., 2020), or general sequentially ordered data (Király and Oberhauser, 2019; Chevyrev and Oberhauser, 2022); a survey of kernels for structured data is provided by Gärtner (2003).

## B  Technical background

In Section B.1 the tensor products $(\bigotimes_{j=1}^{d} \mathcal{H}_j)$ and direct sums of Hilbert spaces $(\bigoplus_{i \in I} \mathcal{H}_i)$ are recalled. Section B.2 is about tensor algebras over Hilbert spaces $(\prod_{m \geq 0} \mathcal{H}^{\otimes m})$.

### B.1  Tensor products and direct sums of Banach and Hilbert spaces

**Tensor products of Hilbert spaces.**    For Hilbert spaces $\mathcal{H}, \ldots, \mathcal{H}_d$ and $(h_1, \ldots, h_d) \in \mathcal{H}_1 \times \cdots \times \mathcal{H}_d$, the multi-linear operator $h_1 \otimes \cdots \otimes h_d \in \mathcal{H}_1 \otimes \cdots \otimes \mathcal{H}_d$ is defined as

$$(h_1 \otimes \cdots \otimes h_d)(f_1, \ldots, f_d) = \prod_{j=1}^{d} \langle h_j, f_j \rangle_{\mathcal{H}_j}$$

for all $(f_1, \ldots, f_d) \in \mathcal{H}_1 \times \cdots \times \mathcal{H}_d$. By extending the inner product

$$\langle a_1 \otimes \cdots \otimes a_d, b_1 \otimes \cdots \otimes b_d \rangle_{\mathcal{H}_1 \otimes \cdots \otimes \mathcal{H}_d} := \prod_{j=1}^{d} \langle a_j, b_j \rangle_{\mathcal{H}_j}$$

to finite linear combinations of $a_1 \otimes \cdots \otimes a_d$-s

$$\left\{ \sum_{i=1}^{n} c_i \otimes_{j=1}^{d} a_{i,j} \ : \ c_i \in \mathbb{R}, a_{i,j} \in \mathcal{H}_j, \ n \geq 1 \right\}$$

by linearity, and taking the topological completion one arrives at $\mathcal{H}_1 \otimes \cdots \otimes \mathcal{H}_d$. Specifically, if $(\mathcal{H}_1, k_1), \ldots, (\mathcal{H}_d, k_d)$ are RKHSs, then so is $\mathcal{H}_1 \otimes \cdots \otimes \mathcal{H}_d = \mathcal{H}_{\otimes_{j=1}^{d} k_j}$ (Berlinet and Thomas-Agnan, 2004, Theorem 13) with the tensor product kernel

$$\left( \otimes_{j=1}^{d} k_j \right) \left( (x_1, \ldots, x_d), (x_1', \ldots, x_d') \right) := \prod_{j=1}^{d} k_j \left( x_j, x_j' \right),$$

where $(x_1, \ldots, x_d), (x_1', \ldots, x_d') \in \mathcal{X}_1 \times \cdots \times \mathcal{X}_d$.

**Tensor products of Banach spaces.**    For Banach spaces $\mathcal{B}_1, \ldots \mathcal{B}_d$, the construction of $\mathcal{B}_1 \otimes \cdots \otimes \mathcal{B}_d$ is a little more involved (Lang, 2002) as one cannot rely on an inner product.

**Direct sums of Hilbert and Banach spaces.**    Let $(\mathcal{H}_i)_{i \in I}$ be Hilbert or Banach spaces where $I$ is some index set. The direct sum of $\mathcal{H}_i$-s— written as $\bigoplus_{i \in I} \mathcal{H}_i$—consists of ordered tuples $h = (h_i)_{i \in I}$ such that $h_i \in \mathcal{H}_i$ for all $i \in I$ and $h_i = 0$ for all but a finite number of $i \in I$. Operations (addition, scalar multiplication) are performed coordinate-wise, and the inner product of $a, b \in \bigoplus_{i \in I} \mathcal{H}_i$ is defined as $\langle a, b \rangle_{\bigoplus_{i \in I} \mathcal{H}_i} = \sum_{i \in I} a_i b_i$.

### B.2  Tensor algebras

The tensor algebra $\mathrm{T}_j$ over a Hilbert space $\mathcal{H}_j$ is defined as the topological completion of the space

$$\bigoplus_{m \geq 0} \mathcal{H}_j^{\otimes m}.$$

Note that it can equivalently be defined as the subset of $(h_0, h_1, h_2, \ldots) \in \prod_{m \geq 0} \mathcal{H}_j^{\otimes m}$ such that $\sum_{m \geq 0} \|h_m\|_{\mathcal{H}_j^{\otimes m}}^2 < \infty$, and as such it is a Hilbert space with norm

$$\|(h_0, h_1, h_2, \ldots)\|_{\prod_{m \geq 0} \mathcal{H}_j^{\otimes m}}^2 = \sum_{m \geq 0} \|h_m\|_{\mathcal{H}_j^{\otimes m}}^2.$$

$\mathrm{T}_j$ is also an algebra, endowed with the tensor product over $\mathcal{H}_j$ as its product. For $a = (a_0, a_1, a_2, a_2 \ldots), b = (b_0, b_1, b_2, b_2 \ldots) \in \mathrm{T}_j$, their product can be written down in coordinates as

$$a \cdot b = \left( \sum_{i=0}^m a_i \otimes b_{m-i} \right)_{m \geq 0}.$$

For a sequence $\mathcal{H}_1, \ldots, \mathcal{H}_d$ of Hilbert spaces, we define

$$\mathrm{T} := \mathrm{T}_1 \otimes \cdots \otimes \mathrm{T}_d,$$

where $\mathrm{T}_j = \prod_{m \geq 0} \mathcal{H}_j^{\otimes m}$ $(j = 1, \ldots, d)$. Let $\mathcal{H} = \mathcal{H}_1 \times \cdots \times \mathcal{H}_d$, and recall that given a tuple of integers $\mathbf{i} = (i_1, \ldots, i_d) \in \mathbb{N}^d$ we define $\mathcal{H}^{\otimes \mathbf{i}} := \mathcal{H}_1^{\otimes i_1} \otimes \cdots \otimes \mathcal{H}_d^{\otimes i_d}$. This allows us to write down a multi-grading for $\mathrm{T}$ as

$$\mathrm{T} = \prod_{\mathbf{i} \in \mathbb{N}^d} \mathcal{H}^{\otimes \mathbf{i}}. \tag{8}$$

Note that this gives credence to us using multi-indices $\mathbf{i} \in \mathbb{N}^d$ to describe elements of the tensor algebra, as the multi-indices form its multi-grading.

Furthermore, $\mathrm{T}$ is a multi-graded algebra when endowed with the (linear extension of the) following multiplication defined on the components of $\mathrm{T}$

$$\star : \mathcal{H}^{\otimes \mathbf{i}^1} \times \mathcal{H}^{\otimes \mathbf{i}^2} \to \mathcal{H}^{\otimes (\mathbf{i}^1 + \mathbf{i}^2)}, \tag{9}$$
$$(x_1 \otimes \cdots \otimes x_d) \star (y_1 \otimes \cdots \otimes y_d) = (x_1 \cdot y_1) \otimes \cdots \otimes (x_d \cdot y_d),$$

so that for $a = \left( a^{\mathbf{i}} \right)_{\mathbf{i} \in \mathbb{N}^d}, b = \left( b^{\mathbf{i}} \right)_{\mathbf{i} \in \mathbb{N}^d} \in \mathrm{T}$, their product can be written down as

$$(a \star b)^{\mathbf{i}} = \sum_{\mathbf{i}^1 + \mathbf{i}^2 = \mathbf{i}} a^{\mathbf{i}^1} \star b^{\mathbf{i}^2} \tag{10}$$

where addition of tuples $\mathbf{i}^1, \mathbf{i}^2 \in \mathbb{N}^d$ is defined as $\mathbf{i}^1 + \mathbf{i}^2 = \left( i_1^1 + i_1^2, \ldots, i_d^1 + i_d^2 \right)$. With the degree of a tuple defined as $\deg(\mathbf{i}) = i_1 + \cdots + i_d$, $\mathrm{T}$ is also a graded algebra, with the grading written down as

$$\mathrm{T} = \prod_{m \geq 0} \bigoplus_{\{\mathbf{i} \in \mathbb{N}^d : \deg(\mathbf{i}) = m\}} \mathcal{H}^{\otimes \mathbf{i}},$$

so that if one multiplies two elements together, the degree of their product is the sum of their degree.

Finally we note that $\mathrm{T}$ is a unital algebra and the unit has the explicit form

$$(1, 0, 0, \ldots),$$

i.e. the element consisting of only a 1 at degree 0.

## C   Proofs

This section is dedicated to proofs. The equivalence between the combinatorial expressions of cumulants and the definition via a moment generating function is proved in Section C.2. The derivation of our main results (Theorem 2 and Theorem 3) are detailed in Section C.3.

## C.1 Equivalent definitions of cumulants in $\mathbb{R}^d$

Here we introduce a classical definition of cumulants via a moment generating function and its equivalence to the combinatorial expressions. If $X = (X_1, \ldots, X_d)$ is an $\mathbb{R}^d$-valued random variable distributed according to $X \sim \gamma$, then recall that

$$\mu^{\mathbf{i}}(\gamma) = \mathbb{E}[X_1^{i_1} \cdots X_d^{i_d}] \in \mathbb{R}$$

for $\mathbf{i} = (i_1, \ldots, i_d) \in \mathbb{N}^d$.

The following definition of the cumulants $\kappa^{\mathbf{i}}(\gamma)$ of $\gamma$ are equivalent

1. $\sum_{\mathbf{i} \in \mathbb{N}^d} \kappa^{\mathbf{i}}(\gamma) \frac{\boldsymbol{\theta}^{\mathbf{i}}}{\mathbf{i}!} = \log \sum_{\mathbf{i} \in \mathbb{N}^d} \mu^{\mathbf{i}}(\gamma) \frac{\boldsymbol{\theta}^{\mathbf{i}}}{\mathbf{i}!}$,

2. $\kappa^{\mathbf{i}}(\gamma) = \sum_{\pi \in P(d)} c_\pi \mu^{\mathbf{i}}(\gamma_\pi^{\mathbf{i}})$,

where $\boldsymbol{\theta} = (\theta_1, \ldots, \theta_d) \in \mathbb{R}^d$, $c_\pi = (-1)^{|\pi|}(|\pi| - 1)!$. The equivalence between these two definitions of cumulants via a generating function and via their combinatorial definition, is classical (McCullagh, 2018) even if our notation here is non-standard in the classical case. This equivalence is also at the heart of many proofs about properties of cumulants since some properties are easier to prove via one or the other definition.

## C.2 Equivalent definitions of cumulants in RKHS

In the main text, we defined cumulants in RKHS by mimicking the combinatorial definition of cumulants in $\mathbb{R}^d$. It is natural and useful to also have the analogous definition via a "generating function" for RKHS-valued random variables. However, to generalize the definition via the logarithm of the moment generating function to random variables in RKHS, requires to define a logarithm for tensor series of moments. In this part, we show that this can be done and that indeed the two definitions are equivalent.

We use the shorthand $\kappa(\gamma) := \kappa_{k_1, \ldots, k_d}(\gamma)$, $\mu(\gamma) := \mu_{k_1, \ldots, k_d}(\gamma)$, and we overload the notation $(X_1, \ldots, X_d)$ with $(k_1(\cdot, X_1), \ldots, k_d(\cdot, X_d))$. With this notation, we show that given coordinates $\mathbf{i} \in \mathbb{N}^d$, one may express the generalized cumulant $\kappa^{\mathbf{i}}(\gamma)$ as either a combinatorial sum over moments indexed by partitions, or by using the cumulant generating function.

More specifically, we show that the generalized cumulant of a probability measure $\gamma$ on $\mathcal{H}_1 \times \cdots \times \mathcal{H}_d$ defined as

$$\kappa^{\mathbf{i}}(\gamma) = \sum_{\pi \in P(m)} c_\pi \mathbb{E}_{\gamma_\pi^{\mathbf{i}}}(X^{\otimes \mathbf{i}}),$$

where $c_\pi = (-1)^{|\pi|-1}(|\pi| - 1)!$ can also be expressed as coordinates in the tensorized logarithm of the moment series. Motivated by the Taylor series expansion of the classic logarithm, we define

$$\log : \mathrm{T} \to \mathrm{T}, \quad x \mapsto \sum_{n \geq 1} \frac{(-1)^{n-1}}{n}(x - 1)^{\star n},$$

where $\star$ denotes the product as defined in (9) and for $t \in \mathrm{T}$, $t^{\star n}$ is defined as

$$t^{\star n} = \underbrace{t \star \cdots \star t}_{n \text{ - times}},$$

or coordinate-wise $(t^{\star n})^{\mathbf{i}} = \sum_{\mathbf{i}^1 + \cdots + \mathbf{i}^n = \mathbf{i}} t^{\mathbf{i}^1} \star \cdots \star t^{\mathbf{i}^n}$ for $\mathbf{i} \in \mathbb{N}^d$. Note that unlike the classical logarithm $\log : \mathbb{R}_+ \to \mathbb{R}$, the tensorized logarithm is defined on the whole space as a formal expression. This can be summarized in the following lemma:

**Lemma 4.**

$$\kappa^{\mathbf{i}}(\gamma) = \sum_{\pi \in P(m)} c_\pi \mathbb{E}_{\gamma_\pi^{\mathbf{i}}}(X^{\otimes \mathbf{i}}) = \left( \log \mu(\gamma^{\mathbf{i}}) \right)^{\mathbf{1}_m}, \tag{11}$$

*where* $\mathbf{1}_m = (1, \ldots, 1) \in \mathbb{N}^m$.

By iterating (10) we can express (11) as

$$\sum_{j=1}^{m} \frac{(-1)^{j-1}}{j} \sum_{\mathbf{i}^1+\cdots+\mathbf{i}^j=\mathbf{1}_m} \mu^{\mathbf{i}^1}(\gamma^{\mathbf{i}}) \star \cdots \star \mu^{\mathbf{i}^j}(\gamma^{\mathbf{i}}),$$

and our goal is to express this as a sum over partitions. We will use the notation $[n] = \{1,\ldots,n\}$. We can achieve our goal in two parts:

1. Show that for a fixed $\mathbf{i} \in \mathbb{N}^d$ with $\deg(\mathbf{i}) = m$ we can express (11) as a sum over all surjective functions from $[m]$ to $[j]$.

2. Show that this sum over functions reduces to a sum over partitions.

**Part 1.** Note that given $\mathbf{i}^1 + \cdots + \mathbf{i}^j = \mathbf{1}_m$ we may define $h : [m] \to [j]$ by the relation $(\mathbf{i}^{h(n)})_n = 1$, that is, we take $h(n)$ to be the index $c$ for which the multi-index $\mathbf{i}^c$ is 1 at $n$. Note that this function is necessarily surjective since the sum is taken over non-zero multi-indices. Equivalently, for any surjective function $h : [m] \to [j]$ we may define multi-indices by setting

$$(\mathbf{i}^c)_n = \begin{cases} 1 & \text{if } n \in h^{-1}(c) \\ 0 & \text{otherwise} \end{cases}.$$

Note that any such multi-index will be non-zero since the function is assumed to be surjective. With this identification we can write

$$\big(\log \mu(\gamma^{\mathbf{i}})\big)^{\mathbf{1}_m} = \sum_{j=1}^{m} \frac{(-1)^{j-1}}{j} \sum_{h:[m]\to[j]} \mu^{\mathbf{i}^{h^{-1}(1)}}(\gamma^{\mathbf{i}}) \star \cdots \star \mu^{\mathbf{i}^{h^{-1}(j)}}(\gamma^{\mathbf{i}}).$$

**Part 2.** Recall that given a function $h : [m] \to [j]$ we can associate it to its corresponding partition $\pi_h \in \mathcal{P}(m)$ by considering the set $\{h^{-1}(1),\ldots,h^{-1}(j)\}$, and there are exactly $j!$ different functions corresponding to a given partition, which are given by re-ordering the values $1,\ldots,j$. This reordering of the blocks does not change the summands since the marginals of the partition measure are always copies of each other and hence self-commute, hence a product of moments like $\mu^{\mathbf{i}^{h^{-1}(1)}}(\gamma^{\mathbf{i}}) \star \cdots \star \mu^{\mathbf{i}^{h^{-1}(j)}}(\gamma^{\mathbf{i}})$ can always be written as $\mu^{\mathbf{i}}(\gamma^{\mathbf{i}}_{\pi_h})$, the $\mathbf{i}$-th coordinate of the moment sequence of the partition measure $\gamma^{\mathbf{i}}_{\pi_h}$. With this in mind we can write

$$\big(\log \mu(\gamma^{\mathbf{i}})\big)^{\mathbf{1}_m} = \sum_{j=1}^{m} \frac{(-1)^{j-1}}{j} \sum_{h:[m]\to[j]} \mu^{\mathbf{i}}(\gamma^{\mathbf{i}}_{\pi_h}) = \sum_{\pi \in P(m)} \frac{(-1)^{|\pi|-1}}{|\pi|} |\pi|! \mu^{\mathbf{i}}(\gamma^{\mathbf{i}}_{\pi})$$

$$= \sum_{\pi \in P(m)} c_\pi \mu^{\mathbf{i}}(\gamma^{\mathbf{i}}_{\pi}) = \sum_{\pi \in P(m)} c_\pi \mathbb{E}_{\gamma^{\mathbf{i}}_{\pi}}(X^{\otimes \mathbf{i}}).$$

From this it immediately follows that for two probability measures $\gamma, \eta$ we can write

$$\langle \kappa^{\mathbf{i}}(\gamma), \kappa^{\mathbf{i}}(\eta) \rangle_{\mathcal{H}^{\otimes \mathbf{i}}} = \langle \sum_{\pi \in P(m)} c_\pi \mathbb{E}_{\gamma^{\mathbf{i}}_{\pi}}(X^{\otimes \mathbf{i}}), \sum_{\tau \in P(m)} c_\tau \mathbb{E}_{\eta^{\mathbf{i}}_{\tau}}(Y^{\otimes \mathbf{i}}) \rangle_{\mathcal{H}^{\otimes \mathbf{i}}}$$

$$= \sum_{\pi,\tau \in P(m)} c_\pi c_\tau \mathbb{E}_{(X,Y)\sim \gamma^{\mathbf{i}}_{\pi} \otimes \eta^{\mathbf{i}}_{\tau}} \langle X^{\otimes \mathbf{i}}, Y^{\otimes \mathbf{i}} \rangle_{\mathcal{H}^{\otimes \mathbf{i}}}.$$

Lemma 1 then follows from the definition of the tensor products.

## C.3 Proof of Theorem 2 and Theorem 3

In this section we present the proofs of Theorem 2 and Theorem 3. We do this in a slightly more abstract setting where the feature maps take values in Banach spaces for clarity, until the end when we again restrict our attention to RKHSs. We start out by showing that polynomial functions of the feature maps characterize measures (Lemma 5). From this result we show that cumulants have the same property (Theorem 4), and lastly that this also holds when working directly with the kernels (Proposition 1).

A *monomial* on separable Banach spaces $\mathcal{B}_1, \ldots, \mathcal{B}_d$ is any expression of the form

$$M(x_1, \ldots, x_d) = \prod_{j=1}^{i_1} \langle f_j^1, x_1 \rangle \cdots \prod_{j=1}^{i_d} \langle f_j^d, x_d \rangle$$

for some $(i_1, \ldots, i_d) \in \mathbb{N}^d$, where $f_j^i \in \mathcal{B}_i^\star$ are elements of the dual space $\mathcal{B}_i^\star$ and $x_i \in \mathcal{B}_i$.[5] Finite linear combinations of monomials are called the *polynomials*. Recall that a set of functions $F$ on a set $S$ is said to *separate the points* of $S$ if for every $x \neq y \in S$ there exists $f \in F$ such that $f(x) \neq f(y)$.

**Lemma 5** (Polynomial functions of feature maps characterize probability measures)**.** *Let $\mathcal{X}_1, \ldots, \mathcal{X}_d$ be Polish spaces, $\mathcal{B}_1 \ldots, \mathcal{B}_d$ separable Banach spaces and $\varphi_i : \mathcal{X}_i \to \mathcal{B}_i$ be continuous, bounded, and injective functions. Then the set of functions on the Borel probability measures $\mathcal{P}\left(\prod_{i=1}^d \mathcal{X}_i\right)$ of $\prod_{i=1}^d \mathcal{X}_i$*

$$\mathcal{P}\left(\prod_{i=1}^d \mathcal{X}_i\right) \to \mathbb{R}, \quad \gamma \mapsto \int_{\prod_{i=1}^d \mathcal{X}_i} p\big(\varphi_1(x_1), \ldots, \varphi_d(x_d)\big) \mathrm{d}\gamma(x_1, \ldots, x_d),$$

*where $p$ ranges over all polynomials, separates the points of $\mathcal{P}\left(\prod_{i=1}^d \mathcal{X}_i\right)$.*

*Proof.* We first show that the pushforward map

$$\prod_{i=1}^d \varphi_i : \mathcal{P}\left(\prod_{i=1}^d \mathcal{X}_i\right) \to \mathcal{P}\left(\prod_{i=1}^d \mathcal{B}_i\right)$$

is injective. This is done in two parts, first we show that every Borel measure on $\prod_{i=1}^d \mathcal{X}_i$ is a Radon measure, then we show that the pushforward map is injective on Radon measures. To see the first part, note that since $\mathcal{X}_1, \ldots, \mathcal{X}_d$ are Polish spaces, so is their product space $\prod_{i=1}^d \mathcal{X}_i$ (Dudley 2004, Theorem 2.5.7; Willard 1970, Theorem 16.4c), and since Borel measures on Polish spaces are Radon measures (Bogachev, 2007, Theorem 7.1.7), any $\gamma \in \mathcal{P}(\prod_{i=1}^d \mathcal{X}_i)$ must be a Radon measure.

For the second part, note that

$$\prod_{i=1}^d \varphi_i : \prod_{i=1}^d \mathcal{X}_i \to \prod_{i=1}^d \mathcal{B}_i, \quad \left(\prod_{i=1}^d \varphi_i\right)(x_1, \ldots, x_d) \mapsto \prod_{i=1}^d \varphi_i(x_i)$$

is a norm bounded, continuous injection. Since $\prod_{i=1}^d \mathcal{B}_i$ is a Hausdorff space, $\prod_{i=1}^d \varphi_i$ is a homeomorphism on compacts since continuous injections into Hausdorff spaces are homeomorphisms on compacts (Rudin, 1953, Theorem 4.17). Let $\mu, \nu \in \mathcal{P}\left(\prod_{i=1}^d \mathcal{X}_i\right)$ be two Radon measures such that their pushforwards are the same $\prod_{i=1}^d \varphi_i(\mu) = \prod_{i=1}^d \varphi_i(\nu)$, then for any compact $C \subseteq \prod_{i=1}^d \mathcal{X}_i$ we have $\mu(C) = \nu(C)$ as $\prod_{i=1}^d \varphi_i : C \to \prod_{i=1}^d \varphi_i(C)$ is a homeomorphism. Since Radon measures are characterized by their values on compacts, this implies that $\mu = \nu$. Hence the pushforward map is injective.

Denote by $K$ the image of $\prod_{i=1}^d \mathcal{X}_i$ under the mapping $\prod_{i=1}^d \varphi_i$ in $\prod_{i=1}^d \mathcal{B}_i$. Note that $K$ is a bounded Polish space. It is enough to show that the polynomials separate the points of $\mathcal{P}(K)$. To see this, note that the polynomials form an algebra of continuous functions that separate the points of $\prod_{i=1}^d \mathcal{B}_i$, and when restricted to $K$ they are bounded, since $K$ is norm bounded. Since $K$ is Polish, any Borel measure is Radon, and we can apply the Stone-Weierstrass theorem for Radon measures (Bogachev, 2007, Exercise 7.14.79) to get the assertion. $\square$

In what follows we will use the following index notation for linear functionals. Fix some tuple $\mathbf{i} = (i_1, \ldots, i_d) \in \mathbb{N}^d$ with $\deg(\mathbf{i}) = m$. Given separable Banach spaces $\mathcal{B}_1 \ldots, \mathcal{B}_d$ we use the notation

$$\mathcal{B}^{\otimes \mathbf{i}} := \mathcal{B}_1^{\otimes i_1} \otimes \cdots \otimes \mathcal{B}_d^{\otimes i_d}$$

---

[5]These monomials naturally extend the classical ones.

and given an element $x = (x_1, \ldots, x_d) \in \prod_{i=1}^d \mathcal{B}_i$ we write $x^{\mathbf{i}} := x_1^{\otimes i_1} \otimes \cdots \otimes x_d^{\otimes i_d}$ so that $x^{\mathbf{i}} \in \mathcal{B}^{\otimes \mathbf{i}}$. If we have functions $(\varphi_i)_{i=1}^d$ such that $\varphi_i : \mathcal{X}_i \to \mathcal{B}_i$ on some Polish spaces $\mathcal{X}_1, \ldots, \mathcal{X}_d$, then we write

$$\varphi^{\otimes \mathbf{i}} := \varphi_1^{\otimes i_1} \otimes \cdots \otimes \varphi_d^{\otimes i_d}, \quad \varphi^{\otimes \mathbf{i}} : \prod_{i=1}^d \mathcal{X}_i \to \mathcal{B}^{\otimes \mathbf{i}}.$$

Given a collection of linear functionals $F \in \prod_{j=1}^d \left(\mathcal{B}_j^\star\right)^{i_j}$ such that $F = (f_1, \ldots, f_d)$ we write

$$F^{\otimes \mathbf{i}} := f_1 \otimes \cdots \otimes f_d, \quad F^{\otimes \mathbf{i}} \in \left(\mathcal{B}^{\otimes \mathbf{i}}\right)^\star.$$

Note the following trick: the monomials on $\prod_{i=1}^d \mathcal{B}_i$ are exactly functions of the form

$$x \mapsto \langle F^{\otimes \mathbf{i}}, x^{\mathbf{i}} \rangle$$

for $F = (f_1, \ldots, f_d)$, this will be used in the proofs. We can now restate and prove the our theorem. Note that the cumulants here are defined like in Definition 4 which is a sensible definition even if the feature maps are not associated to kernels.

**Theorem 4** (Generalization of Theorem 2 and Theorem 3). *Let $\mathcal{X}_1, \ldots, \mathcal{X}_d$ be Polish spaces and $\varphi_i : \mathcal{X}_i \to \mathcal{B}_i$ be continuous, bounded and injective feature maps into separable Banach spaces $\mathcal{B}_i$ for $i = 1, \ldots d$. Let $\gamma$ and $\eta$ be probability measures on $\mathcal{X}_1 \times \cdots \times \mathcal{X}_d$. Then*

*1. $\gamma = \eta$ if and only if $\kappa(\gamma) = \kappa(\eta)$.*

*2. $\gamma = \bigotimes_{i=1}^d \gamma|_{\mathcal{X}_i}$ if and only if the cross cumulants vanish, that is $\kappa^{\mathbf{i}}(\gamma) = 0$ for all $\mathbf{i} \in \mathbb{N}_+^d$.*

*Proof.*
● Item 2: We want to show that the cross cumulants vanish if and only if $\gamma = \bigotimes_{i=1}^d \gamma|_{\mathcal{X}_i}$. By Lemma 5 it is enough to show that

$$\mathbb{E}_\gamma \Big[ p\big(\varphi_1(X_1), \ldots, \varphi_d(X_d)\big) \Big] = \mathbb{E}_{\bigotimes_{i=1}^d \gamma|_{\mathcal{X}_i}} \Big[ p\big(\varphi_1(X_1), \ldots, \varphi_d(X_d)\big) \Big]$$

for any monomial function $p$. Let us take linear functionals $F = (f_1, \ldots, f_d)$ and note that

$$\langle F^{\mathbf{i}}, \kappa^{\mathbf{i}}(\gamma) \rangle = \sum_{\pi \in P(d)} c_\pi \mathbb{E}_{\gamma_\pi^{\mathbf{i}}} \big[ f_1(\varphi_1(X_1)) \cdots f_d(\varphi_d(X_d)) \big]$$

which is the classical cumulant of the vector-valued random variable

$$\big((f_1 \circ \varphi_1)(X_1), \ldots, (f_d \circ \varphi_d)(X_d)\big),$$

where $(X_1, \ldots, X_d) \sim \gamma$. Hence by classical results (Speed, 1983), all cross cumulants of $\big((f_1 \circ \varphi_1)(X_1), \ldots, (f_d \circ \varphi_d)(X_d)\big)$ vanish if and only if the cross moments split, that is to say

$$\mathbb{E}_\gamma \Big[ p\big((f_1 \circ \varphi_1)(X_1), \ldots, (f_d \circ \varphi_d)(X_d)\big) \Big] = \mathbb{E}_{\bigotimes_{i=1}^d \gamma|_{\mathcal{X}_i}} \Big[ p\big((f_1 \circ \varphi_1)(X_1), \ldots, (f_d \circ \varphi_d)(X_d)\big) \Big]$$

for any monomial $p$ on $\mathbb{R}^d$. Since $f_1, \ldots, f_d$ were arbitrary this holds for all monomials, which shows the assertion.

● Item 1: By assumption $\kappa^{\mathbf{i}}(\gamma) = \kappa^{\mathbf{i}}(\eta)$ for every $\mathbf{i} \in \mathbb{N}^d$; this implies that $\mathbb{E}_\gamma p(\varphi_1, \ldots, \varphi_d) = \mathbb{E}_\eta p(\varphi_1, \ldots, \varphi_d)$ for any polynomial $p$, so we can apply Lemma 5. □

**Proposition 1** (Theorem 2 and Theorem 3). *Let $\mathcal{X}_1, \ldots, \mathcal{X}_d$ be Polish spaces and $k_i : \mathcal{X}_i^2 \to \mathbb{R}$ be a collection of bounded, continuous, point-separating kernels. Let $\gamma$ and $\eta$ be be probability measures on $\mathcal{X}_1 \times \cdots \times \mathcal{X}_d$. Then*

*1. $\gamma = \eta$ if and only if $\kappa_{k_1, \ldots, k_d}(\gamma) = \kappa_{k_1, \ldots, k_d}(\eta)$.*

*2. $\gamma = \bigotimes_{i=1}^d \gamma|_{\mathcal{X}_i}$ if and only if $\kappa_{k_1, \ldots, k_d}^{\mathbf{i}}(\gamma) = 0$ for all $\mathbf{i} \in \mathbb{N}_+^d$.*

*Proof.* We reduce the proof to the checking of the conditions of Theorem 4. Let $\varphi_i$ denote the canonical feature map of the kernel $k_i$, and let $\mathcal{B}_i := \mathcal{H}_{k_i}$ be the RKHS associated to $k_i$ ($i \in \{1, \ldots, d\}$). For all $i \in \{1, \ldots, d\}$, $\varphi_i$ is (i) bounded by the boundedness of $k_i$ since $\|\varphi_i(x)\|_{\mathcal{H}_{k_i}}^2 = k_i(x, x) \leq \sup_{x \in \mathcal{X}_i} |k_i(x, x)| < \infty$, (ii) continuous by the continuity of $k_i$ (Steinwart and Christmann, 2008, Lemma 4.29), (iii) injective by the point-separating property of $k_i$. The separability of $\mathcal{H}_{k_i}$ follows (Steinwart and Christmann, 2008, Lemma 4.33) from the separability of $\mathcal{X}_i$ and the continuity of $k_i$ ($i \in \{1, \ldots, d\}$). Note: Details on the expected kernel trick part of Theorem 2 and Theorem 3 are provided in Section E. $\qquad\square$

# D    Additional experiments and details

Here we give additional details on the experiments that were performed, and discuss some further experiments that did not fit into the main text.

**Background on permutation testing.**    Permutation testing works by bootstrapping the distribution of a test statistic under the null hypothesis. This allows the user to estimate confidence intervals under the null, which is a powerful all-purpose way of doing so when analytic expressions are unavailable. As an example, assume we have two probability measures $\gamma, \eta$ on $\mathcal{X}$ with i.i.d. samples $x_1, \ldots, x_N \sim \gamma, y_1, \ldots, y_N \sim \eta$. If the null hypothesis is that $\gamma = \eta$ then we may set

$$(z_1, \ldots, z_{2N}) := (x_1, \ldots, x_N, y_1, \ldots, y_N)$$

so that for any permutation $\sigma$ on $2N$ elements, we get two different set of of i.i.d. samples from $\gamma = \eta$ by using the empirical measures

$$\tilde{\gamma}_\sigma := (z_{\sigma(1)}, \ldots, z_{\sigma(N)}), \quad \tilde{\eta}_\sigma := (z_{\sigma(N+1)}, \ldots, z_{\sigma(2N)})$$

and for any statistic $S : \mathcal{P}(\mathcal{X})^2 \to \mathbb{R}$, we may estimate $S(\gamma, \eta)$ under the null by sampling from $S(\tilde{\gamma}_\sigma, \tilde{\eta}_\sigma)$. If the null hypothesis were true, we might expect $S(\gamma, \eta)$ to lie in a region with high probability of the permutation estimator, and we can use this as a criteria for rejecting the null. Under fairly weak assumptions, this yields a test at the appropriate level (Chung and Romano, 2013).

**Comparing a uniform and a mixture.**    Any uniform random variable over a symmetric interval will have 0 mean and skewness, so a symmetric mixture only needs to match the variance. If $X$ is a $50/50$ mixture of $U[a, b]$ and $U[-a, -b]$ then

$$\mathrm{Var}(X) = \frac{2}{3} \left( b^2 + ba + a^2 \right)$$

so if $Y$ is distributed according to $U[-c, c]$ then we only need to solve

$$b^2 + ba + a^2 = c^2$$

which is straightforward for a given $a$ and $c$.

**Computational complexity of estimators.**    The V-statistic for $d^{(2)}$ as written in Lemma 2 is bottlenecked by the matrix multiplications. We may note however that for two matrices $\mathbf{A}, \mathbf{B}$ it holds that

$$\mathrm{Tr}(\mathbf{A}^\top \mathbf{B}) = \langle \mathbf{A} \circ \mathbf{B} \rangle,$$

where $\langle \cdot \rangle$ denotes the sum over elements and $\circ$ denotes the Hadamard product. We also note that for for $\mathbf{H}_n = \frac{1}{n} \mathbf{1}_n \mathbf{1}_n^\top$ we have $(\mathbf{A}\mathbf{H}_n)_{i,j} = \frac{1}{n} \sum_{c=1}^n A_{i,c}$. Using both of these tricks we may compute both $d^{(2)}$ and CSIC without any matrix multiplications, which brings the computational complexity down to $O(N^2)$ for both. For a comparison of actual computation time, see Fig. 6 and Fig. 7, where the average computational times for out methods are compared to the KME and and HSIC for $N$ between 50 and 2000.

**Type I error on the Seoul Bicycle data.**    The results when comparing the winter data to itself is presented in Fig. 8. As we see the performance is similar for both estimators and lies between 5 and 10%.

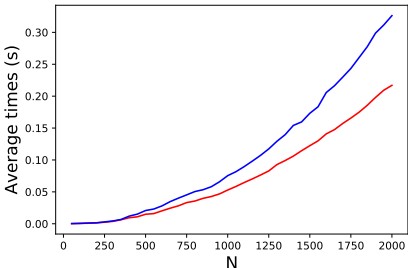

Figure 6: Average computational time in seconds for KME (red) and $d^{(2)}$ (blue) for sample size $N$ between 50 and 2000.

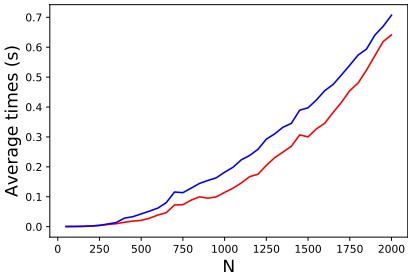

Figure 7: Average computational time in seconds for HSIC (red) and CSIC (blue) for sample size $N$ between 50 and 2000.

**Classical vs. kernelized cumulants.**    Using the same distributions as in the synthetic independence testing experiment, we now compare $X$ with $Y_{0.5}^2$ to contrast independence testing with classical cumulants with their kernelized counterpart. The results are summarized in Table 1 where they are displayed as the median value $\pm$ half the difference between the 75th and 25th percentile. We consider every combination of classical vs. kernelized, variance vs. skewness, and two different sample sizes. One can observe that the classical variance based test performs poorly compared to a classical skewness test, the kernelized variance test is almost as powerful as the kernelized skewness test, and in all cases the kernelized tests deliver higher power.

## E    Kernel trick computations

Here we show how to arrive at the expressions used for the V-statistics used in the experiments.

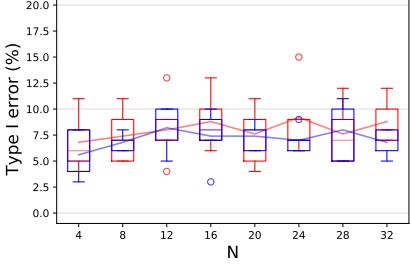

Figure 8: Type I errors using MMD (red) and $d^{(2)}$ (blue) on the Seoul bicycle data set.

Table 1: Comparison of classical and kernelized cumulants for independence testing with both variance and skewness.

| N=20 | Variance | Skewness | N=30 | Variance | Skewness |
|---|---|---|---|---|---|
| Classical | $19\% \pm 3.0\%$ | $56\% \pm 3.5\%$ | Classical | $17\% \pm 0.5\%$ | $68\% \pm 1.0\%$ |
| Rbf kernel | $39\% \pm 4.5\%$ | $59\% \pm 3.0\%$ | Rbf kernel | $65\% \pm 3.5\%$ | $79\% \pm 1.5\%$ |

Given a real analytic function $f(x, \ldots, x_d) = \sum_{\mathbf{i} \in \mathbb{N}^d} f_{\mathbf{i}} x^{\mathbf{i}}$ in $m$ variables with nonzero radius of convergence and Hilbert spaces $\mathcal{H}_1, \ldots, \mathcal{H}_d$ we may (formally) extend $f$ to a function

$$f_\otimes : \prod_{i=1}^d \mathcal{H}_i \to \mathrm{T}, \quad f_\otimes(x_1, \ldots, x_d) = \prod_{\mathbf{i} \in \mathbb{N}^d} f_{\mathbf{i}} x^{\otimes \mathbf{i}}.$$

Moreover, if the Hilbert spaces are RKHSs then we have the following result.

**Lemma 6** (Nonlinear kernel trick)**.** *For any collection of RKHSs $\mathcal{H}_1, \ldots, \mathcal{H}_d$ with feature maps $\varphi_i : \mathcal{X}_i \to \mathcal{H}_i$, assume that $f$ and $g$ are real analytic functions with radii of convergence $r(f)$ and $r(g)$ such that $\max_{1 \le i \le d} \sup_{x \in \mathcal{X}_i} |\varphi_i(x)| < \min(r(f), r(g))$. Then*

$$\langle f_\otimes\big(\varphi_1(x_1), \ldots, \varphi_d(x_d)\big), g_\otimes\big(\varphi_1(y_1), \ldots, \varphi_d(y_d)\big)\rangle_\mathrm{T} = \sum_{\mathbf{i} \in \mathbb{N}^d} f_{\mathbf{i}} g_{\mathbf{i}} k_1(x_1, y_1)^{i_1} \ldots k_d(x_d, y_d)^{i_d}.$$

*Proof.* Since the image of the $\varphi_i$s lie inside the radius of convergence of $f_\otimes$ and $g_\otimes$ the power series converge absolutely and we can write

$$\langle f_\otimes\big(\varphi^{\otimes \mathbf{i}}(x^{\mathbf{i}})\big), g_\otimes\big(\varphi^{\otimes \mathbf{i}}(y^{\mathbf{i}})\big)\rangle_\mathrm{T} = \langle \sum_{\mathbf{i} \in \mathbb{N}^d} f_{\mathbf{i}} \varphi^{\otimes \mathbf{i}}(x^{\mathbf{i}}), \sum_{\mathbf{i} \in \mathbb{N}^d} g_{\mathbf{i}} \varphi^{\otimes \mathbf{i}}(y^{\mathbf{i}})\rangle_\mathrm{T}$$

$$= \sum_{\mathbf{i} \in \mathbb{N}^d} f_{\mathbf{i}} g_{\mathbf{i}} \langle \varphi^{\otimes \mathbf{i}}(x^{\mathbf{i}}), \varphi^{\otimes \mathbf{i}}(y^{\mathbf{i}})\rangle_{\mathcal{H}^{\otimes \mathbf{i}}} = \sum_{\mathbf{i} \in \mathbb{N}^d} f_{\mathbf{i}} g_{\mathbf{i}} k_1(x_1, y_1)^{i_1} \ldots k_d(x_d, y_d)^{i_d},$$

where $\mathcal{H} = \mathcal{H}_1 \times \cdots \times \mathcal{H}_d$. $\qquad\square$

Using Lemma 6, we can choose kernels $k_i : \mathcal{X}_i^2 \to \mathbb{R}$ with associated RKHSs $\mathcal{H}_i$ and feature maps $\varphi_i$ and some $\mathbf{i} \in \mathbb{N}^d$ with $\deg(\mathbf{i}) = m$. We make the observation that with $X = (X_1, \ldots, X_d) \sim \gamma$, $Y = (Y_1, \ldots, Y_d) \sim \eta$ and $k^{\otimes \mathbf{i}}$ and $\mathcal{H}^{\otimes \mathbf{i}}$ as in (4), one has

$$\langle \kappa^{\mathbf{i}}(\gamma), \kappa^{\mathbf{i}}(\eta)\rangle_{\mathcal{H}^{\otimes \mathbf{i}}} = \langle \sum_{\pi \in P(m)} c_\pi \mathbb{E}_{\gamma_\pi^{\mathbf{i}}} \varphi^{\otimes \mathbf{i}}(X^{\mathbf{i}}), \sum_{\tau \in P(m)} c_\tau \mathbb{E}_{\eta_\tau^{\mathbf{i}}} \varphi^{\otimes \mathbf{i}}(Y^{\mathbf{i}})\rangle_{\mathcal{H}^{\otimes \mathbf{i}}}$$

$$= \sum_{\pi, \tau \in P(m)} c_\pi c_\tau \langle \mathbb{E}_{\gamma_\pi^{\mathbf{i}}} \varphi^{\otimes \mathbf{i}}(X^{\mathbf{i}}), \mathbb{E}_{\eta_\tau^{\mathbf{i}}} \varphi^{\otimes \mathbf{i}}(Y^{\mathbf{i}})\rangle_{\mathcal{H}^{\otimes \mathbf{i}}}$$

$$= \sum_{\pi, \tau \in P(m)} c_\pi c_\tau \mathbb{E}_{\gamma_\pi^{\mathbf{i}} \otimes \eta_\tau^{\mathbf{i}}} \langle \varphi^{\otimes \mathbf{i}}(X^{\mathbf{i}}), \varphi^{\otimes \mathbf{i}}(Y^{\mathbf{i}})\rangle_{\mathcal{H}^{\otimes \mathbf{i}}}$$

$$= \sum_{\pi, \tau \in P(m)} c_\pi c_\tau \mathbb{E}_{\gamma_\pi^{\mathbf{i}} \otimes \eta_\tau^{\mathbf{i}}} k^{\otimes \mathbf{i}}((X_1, \ldots, X_m), (Y_1, \ldots, Y_m)),$$

Since

$$\|\kappa^{\mathbf{i}}(\gamma)\|^2_{\mathcal{H}^{\otimes \mathbf{i}}} = \langle \kappa^{\mathbf{i}}(\gamma), \kappa^{\mathbf{i}}(\gamma)\rangle_{\mathcal{H}^{\otimes \mathbf{i}}}$$

$$\|\kappa^{\mathbf{i}}(\gamma) - \kappa^{\mathbf{i}}(\eta)\|^2_{\mathcal{H}^{\otimes \mathbf{i}}} = \langle \kappa^{\mathbf{i}}(\gamma), \kappa^{\mathbf{i}}(\gamma)\rangle_{\mathcal{H}^{\otimes \mathbf{i}}} + \langle \kappa^{\mathbf{i}}(\eta), \kappa^{\mathbf{i}}(\eta)\rangle_{\mathcal{H}^{\otimes \mathbf{i}}} - 2\langle \kappa^{\mathbf{i}}(\gamma), \kappa^{\mathbf{i}}(\eta)\rangle_{\mathcal{H}^{\otimes \mathbf{i}}}$$

one gets the expected kernel trick statements of Theorem 2 and Theorem 3.

We are now interested in explicitly computing the expression $\|\kappa_{k,\ell}^{(1,2)}(\gamma)\|^2_{\mathcal{H}_k^{\otimes 1} \otimes \mathcal{H}_\ell^{\otimes 2}}$, $\|\kappa_k^{(2)}(\gamma) - \kappa_k^{(2)}(\eta)\|^2_{\mathcal{H}^{(1,1)}}$ and $\|\kappa_k^{(3)}(\gamma) - \kappa_k^{(3)}(\eta)\|^2_{\mathcal{H}_k^{\otimes 3}}$, and their corresponding V-statistics. Recall that for

a (w.l.o.g.) symmetric, measurable function $h(z_1, \ldots, z_m)$, the V-statistic of $h$ with $N$ samples $Z_1, \ldots, Z_N$ is defined as

$$V(h; Z_1, \ldots, Z_N) := N^{-m} \sum_{i_1, \ldots, i_m = 1}^{N} h(Z_{i_1}, \ldots, Z_{i_m}).$$

Under fairly general conditions, the V-statistic converges in distribution to $\mathbb{E}[h(Z_1, \ldots, Z_m)]$ and a well-developed theory describes this convergence (Van der Waart, 2000; Serfling, 1980; Arcones and Giné, 1992).

**Example E.1** (Estimating $\|\kappa_k^{(2)}(\gamma) - \kappa_k^{(2)}(\eta)\|_{\mathcal{H}^{(1,1)}}^2$). *Let $X, X', X'', X'''$ denote independent copies of $\gamma$ and $Y, Y', Y'', Y'''$ denote independent copies of $\eta$. The full expression for $\|\kappa_k^{(2)}(\gamma) - \kappa_k^{(2)}(\eta)\|_{\mathcal{H}^{(1,1)}}^2$ is*

$$
\begin{aligned}
\|\kappa_k^{(2)}(\gamma) - \kappa_k^{(2)}(\eta)\|_{\mathcal{H}^{(1,1)}}^2 = {} & \mathbb{E}k(X, X')k(X'', X''') + \mathbb{E}k(Y, Y')k(Y'', Y''') \\
& + \mathbb{E}k(X, X')^2 + \mathbb{E}k(Y, Y')^2 \\
& + 2\mathbb{E}k(X, Y)k(X', Y) + 2\mathbb{E}k(X, Y)k(X, Y') \\
& - 2\mathbb{E}k(X, Y)k(X', Y') - 2\mathbb{E}k(X, Y)^2 \\
& - 2\mathbb{E}k(X, X')k(X, X'') - 2\mathbb{E}k(Y, Y')k(Y, Y'').
\end{aligned} \tag{12}
$$

*Given samples $(x_i)_{i=1}^N$, $(y_i)_{i=1}^M$ from $\gamma$ and $\eta$ respectively the corresponding V statistic is*

$$
\begin{aligned}
& \frac{1}{N^4} \sum_{i,j,\kappa,l=1}^{N} k(x_i, x_j)k(x_\kappa, x_l) + \frac{1}{M^4} \sum_{i,j,\kappa,l=1}^{M} k(y_i, y_j)k(y_\kappa, y_l) \\
& + \frac{1}{N^2} \sum_{i,j=1}^{N} k(x_i, x_j)^2 + \frac{1}{M^2} \sum_{i,j=1}^{M} k(y_i, y_j)^2 \\
& + \frac{2}{N^2 M} \sum_{i,\kappa=1}^{N} \sum_{j=1}^{M} k(x_i, y_j)k(x_\kappa, y_j) + \frac{2}{N M^2} \sum_{i=1}^{N} \sum_{j,\kappa=1}^{M} k(x_i, y_j)k(x_i, y_\kappa) \\
& - \frac{2}{N^2 M^2} \sum_{i,l=1}^{N} \sum_{j,\kappa=1}^{M} k(x_i, y_j)k(x_\kappa, y_l) - \frac{2}{NM} \sum_{i=1}^{N} \sum_{j=1}^{M} k(x_i, y_j)^2 \\
& - \frac{2}{N^3} \sum_{i,j,\kappa=1}^{N} k(x_i, x_j)k(x_i, x_\kappa) - \frac{2}{M^3} \sum_{i,j,\kappa=1}^{M} k(y_i, y_j)k(y_i, y_\kappa).
\end{aligned} \tag{13}
$$

*Let us define the Gram matrices $\mathbf{K}_x = [k(x_i, x_j)]_{i,j=1}^N \in \mathbb{R}^{N \times N}$, $\mathbf{K}_y = [k(y_i, y_j)]_{i,j=1}^M \in \mathbb{R}^{M \times M}$, $\mathbf{K}_{x,y} = [k(x_i, y_j)]_{i,j=1}^{N,M}$ and let $\mathbf{H}_N = \frac{1}{N} \mathbf{1}_N \mathbf{1}_N^\top \in \mathbb{R}^{N \times N}$, $\mathbf{H}_M = \frac{1}{M} \mathbf{1}_M \mathbf{1}_M^\top \in \mathbb{R}^{M \times M}$ be the centering, then* (13) *can be rewritten as*

$$
\begin{aligned}
& \frac{1}{N^2} \mathrm{Tr}(\mathbf{H}_N \mathbf{K}_x \mathbf{H}_N \mathbf{K}_x) + \frac{1}{M^2} \mathrm{Tr}(\mathbf{H}_M \mathbf{K}_y \mathbf{H}_M \mathbf{K}_y) + \frac{1}{N^2} \mathrm{Tr}(\mathbf{K}_x^2) + \frac{1}{M^2} \mathrm{Tr}(\mathbf{K}_y^2) \\
& + \frac{2}{NM} \mathrm{Tr}(\mathbf{K}_{xy} \mathbf{H}_N \mathbf{K}_{xy}) + \frac{2}{NM} \mathrm{Tr}(\mathbf{K}_{xy} \mathbf{H}_M \mathbf{K}_{xy}^\top) - \frac{2}{NM} \mathrm{Tr}(\mathbf{H}_M \mathbf{K}_{xy}^\top \mathbf{H}_N \mathbf{K}_{xy}) - \frac{2}{NM} \mathrm{Tr}(\mathbf{K}_{xy}^2) \\
& - \frac{2}{N^2} \mathrm{Tr}(\mathbf{K}_x \mathbf{H}_N \mathbf{K}_x) - \frac{2}{M^2} \mathrm{Tr}(\mathbf{K}_y \mathbf{H}_M \mathbf{K}_y)
\end{aligned}
$$

*which simplifies to*

$$
\frac{1}{N^2} \mathrm{Tr}\left[(\mathbf{K}_x(\mathbf{I} - \mathbf{H}_N))^2\right] + \frac{1}{M^2} \mathrm{Tr}\left[(\mathbf{K}_y(\mathbf{I} - \mathbf{H}_M))^2\right] - \frac{2}{NM} \mathrm{Tr}\left[\mathbf{K}_{xy}(\mathbf{I} - \mathbf{H}_M)\mathbf{K}_{xy}^\top(\mathbf{I} - \mathbf{H}_N)\right].
$$

*This estimator can be computed in quadratic time.*

**Example E.2** (Estimating $\|\kappa_{k,\ell}^{(1,2)}(\gamma)\|_{\mathcal{H}_k^{\otimes 1} \otimes \mathcal{H}_\ell^{\otimes 2}}^2$). *Let $k$ denote the kernel on $\mathcal{X}_1$ and $\ell$ denote the kernel on $\mathcal{X}_2$. Let $(X, Y), (X', Y'), (X'', Y''), (X^{(3)}, Y^{(3)}), (X^{(4)}, Y^{(4)}), (X^{(5)}, Y^{(5)})$ denote in-*

*dependent copies of $\gamma \in \mathcal{P}(\mathcal{X}_1 \times \mathcal{X}_2)$. The full expression for $\|\kappa_{k,\ell}^{(1,2)}(\gamma)\|_{\mathcal{H}_k^{\otimes 1} \otimes \mathcal{H}_\ell^{\otimes 2}}^2$ is*

$$
\begin{aligned}
&\mathbb{E}k(X,X')k(X,X')\ell(Y,Y') - 4\mathbb{E}k(X,X')k(X,X'')\ell(Y,Y') \\
&- 2\mathbb{E}k(X,X')k(X,X')\ell(Y,Y'') + 4\mathbb{E}k(X,X')k(X,X'')\ell(Y,Y^{(3)}) \\
&+ 2\mathbb{E}k(X,X')k(X'',X^{(3)})\ell(Y,Y') + 2\mathbb{E}k(X,X')k(X'',X^{(3)})\ell(Y,Y^{(3)}) \\
&+ 4\mathbb{E}k(X,X')k(X'',X')\ell(Y,Y^{(3)}) + \mathbb{E}k(X,X')k(X,X')\ell(Y'',Y^{(3)}) \\
&- 8\mathbb{E}k(X,X')k(X'',X^{(3)})\ell(Y^{(4)},Y') - 4\mathbb{E}k(X,X')k(X'',X')\ell(Y^{(4)},Y^{(3)}) \\
&+ 4\mathbb{E}k(X,X')k(X'',X^{(3)})\ell(Y^{(4)},Y^{(5)}).
\end{aligned}
$$

*Given samples $(x_i, y_i)_{i=1}^N$ from $\gamma$ the corresponding V-statistic for this expression is*

$$
\begin{aligned}
&\frac{1}{N^2}\sum_{i,j=1}^N k(x_i,x_j)k(x_i,x_j)\ell(y_i,y_j) - \frac{4}{N^3}\sum_{i,j,\kappa=1}^N k(x_i,x_j)k(x_i,x_\kappa)\ell(y_i,y_j) \\
&- \frac{2}{N^3}\sum_{i,j,\kappa=1}^N k(x_i,x_j)k(x_i,x_j)\ell(y_i,y_\kappa) + \frac{4}{N^4}\sum_{i,j,\kappa,l=1}^N k(x_i,x_j)k(x_i,x_\kappa)\ell(y_i,y_l) \\
&+ \frac{2}{N^4}\sum_{i,j,\kappa,l=1}^N k(x_i,x_j)k(x_\kappa,x_l)\ell(y_i,y_j) + \frac{2}{N^4}\sum_{i,j,\kappa,l=1}^N k(x_i,x_j)k(x_\kappa,x_l)\ell(y_i,y_l) \\
&+ \frac{4}{N^4}\sum_{i,j,\kappa,l=1}^N k(x_i,x_j)k(x_\kappa,x_j)\ell(y_i,y_l) + \frac{1}{N^4}\sum_{i,j,\kappa,l=1}^N k(x_i,x_j)k(x_i,x_j)\ell(y_\kappa,y_l) \\
&- \frac{8}{N^5}\sum_{i,j,\kappa,l,m=1}^N k(x_i,x_j)k(x_\kappa,x_l)\ell(y_m,y_j) - \frac{4}{N^5}\sum_{i,j,\kappa,l,m=1}^N k(x_i,x_j)k(x_\kappa,x_j)\ell(y_m,y_l) \\
&+ \frac{4}{N^6}\sum_{i,j,\kappa,l,m,n=1}^N k(x_i,x_j)k(x_\kappa,x_l)\ell(y_m,y_n).
\end{aligned}
$$

*Using the shorthand notation $\mathbf{K} = \mathbf{K}_x, \mathbf{L} = \mathbf{L}_y$ and $\mathbf{H} = \mathbf{H}_N$ and denoting by $\circ$ the Hadamard product $[\mathbf{A} \circ \mathbf{B}]_{i,j} = A_{i,j}B_{i,j}$ and $\langle \cdot \rangle$ the sum over all elements of a matrix $\langle \mathbf{A} \rangle = \sum_{i,j=1}^N A_{i,j}$, the V-statistic above can be written in the simpler form*

$$
\begin{aligned}
\frac{1}{N^2}\Big\langle &\mathbf{K} \circ \mathbf{K} \circ \mathbf{L} - 4\mathbf{K} \circ \mathbf{K}\mathbf{H} \circ \mathbf{L} - 2\mathbf{K} \circ \mathbf{K} \circ \mathbf{L}\mathbf{H} \\
&+ 4\mathbf{K}\mathbf{H} \circ \mathbf{K} \circ \mathbf{L}\mathbf{H} + 2\mathbf{K} \circ \mathbf{L}\Big\langle \frac{\mathbf{K}}{N^2} \Big\rangle + 2\mathbf{K}\mathbf{H} \circ \mathbf{H}\mathbf{K} \circ \mathbf{L} \\
&+ 4\mathbf{K} \circ \mathbf{H}\mathbf{K} \circ \mathbf{L}\mathbf{H} + \mathbf{K} \circ \mathbf{K}\Big\langle \frac{\mathbf{L}}{N^2}\Big\rangle - 8\mathbf{K} \circ \mathbf{L}\mathbf{H}\Big\langle \frac{\mathbf{K}}{N^2}\Big\rangle \\
&- 4\mathbf{K} \circ \mathbf{H}\mathbf{K}\Big\langle \frac{\mathbf{L}}{N^2}\Big\rangle + 4\Big\langle \frac{\mathbf{K}}{N^2}\Big\rangle^2 \mathbf{L}\Big\rangle.
\end{aligned}
$$

Again this estimator can be computed in quadratic time.

**Example E.3** (Estimating $\|\kappa_k^{(3)}(\gamma) - \kappa_k^{(3)}(\eta)\|_{\mathcal{H}_k^{\otimes 3}}^2$). *In order to estimate $d^{(3)}(\gamma, \eta)$ we note that one can write*

$$
\begin{aligned}
\|\kappa_k^{(3)}(\gamma) - \kappa_k^{(3)}(\eta)\|_{\mathcal{H}_k^{\otimes 3}}^2 = &\|\kappa_k^{(3)}(\gamma)\|_{\mathcal{H}_k^{\otimes 3}}^2 + \|\kappa_k^{(3)}(\eta)\|_{\mathcal{H}_k^{\otimes 3}}^2 \\
&- 2\langle \kappa_k^{(3)}(\gamma), \kappa_k^{(3)}(\eta)\rangle_{\mathcal{H}_k^{\otimes 3}}.
\end{aligned}
$$

We can estimate the first two terms like in Example E.2, and the third term can be expressed as

$$\langle \kappa_k^{(3)}(\gamma), \kappa_k^{(3)}(\eta) \rangle_{\mathcal{H}_k^{\otimes 3}} = \mathbb{E}k(X,Y)^3 - 3\mathbb{E}k(X,Y)^2 k(X,Y')^2$$

$$- 3\mathbb{E}k(X,Y)^2 k(X',Y)^2 + 6\mathbb{E}k(X,Y)k(X,Y')k(X',Y)$$

$$+ 3\mathbb{E}k(X,Y)^2 k(X',Y') + 2\mathbb{E}k(X,Y)k(X',Y)k(X'',Y)$$

$$+ 2\mathbb{E}k(X,Y)k(X,Y')k(X,Y'') - 6\mathbb{E}k(X,Y)k(X,Y')k(X',Y'')$$

$$- 6\mathbb{E}k(X,Y)k(X',Y)k(X'',Y') + 4\mathbb{E}k(X,Y)k(X',Y')k(X'',Y'').$$

For simplicity we will assume that we have an equal number of samples $(N)$ from both measures $(x_i)_{i=1}^N \in \gamma$ and $(y_i)_{i=1}^N \in \eta$. The V-statistic for $\langle \kappa_k^{(3)}(\gamma), \kappa_k^{(3)}(\eta) \rangle_{\mathcal{H}_k^{\otimes 3}}$ can be expressed as

$$\frac{1}{N^2}\sum_{i,j=1}^N k(x_i,y_j)^3 - \frac{3}{N^3}\sum_{i,j,\kappa=1}^N k(x_i,y_j)^2 k(x_i,y_\kappa)$$

$$- \frac{3}{N^3}\sum_{i,j,\kappa=1}^N k(x_i,y_j)^2 k(x_\kappa,y_i) + \frac{6}{N^4}\sum_{i,j,\kappa,l=1}^N k(x_i,y_j)k(x_i,y_\kappa)k(x_l,y_j)$$

$$+ \frac{3}{N^4}\sum_{i,j,\kappa,l=1}^N k(x_i,y_j)^2 k(x_\kappa,y_l) + \frac{2}{N^4}\sum_{i,j,\kappa,l=1}^N k(x_i,y_j)k(x_\kappa,y_j)k(x_l,y_j)$$

$$+ \frac{2}{N^4}\sum_{i,j,\kappa,l=1}^N k(x_i,y_j)k(x_i,y_\kappa)k(x_i,y_l) - \frac{6}{N^5}\sum_{i,j,\kappa,l,m=1}^N k(x_i,y_j)k(x_i,y_\kappa)k(x_l,y_m)$$

$$- \frac{6}{N^5}\sum_{i,j,\kappa,l,m=1}^N k(x_i,y_j)k(x_\kappa,y_j)k(x_l,y_m) + \frac{4}{N^6}\sum_{i,j,\kappa,l,m,n=1}^N k(x_i,y_j)k(x_\kappa,y_l)k(x_m,y_n).$$

Using the notation $\mathbf{K}_{xy} = [k(x_i,y_j)]_{i,j=1}^N$, this estimator simplifies to

$$\frac{1}{N^2}\bigg\langle \mathbf{K}_{xy} \circ \mathbf{K}_{xy} \circ \mathbf{K}_{xy} - 3\mathbf{K}_{xy} \circ \mathbf{K}_{xy} \circ \mathbf{H}\mathbf{K}_{xy}$$

$$- 3\mathbf{K}_{xy} \circ \mathbf{K}_{xy} \circ \mathbf{K}_{xy}\mathbf{H} + 6\mathbf{K}_{xy} \circ \mathbf{K}_{xy}\mathbf{H} \circ \mathbf{H}\mathbf{K}_{xy}$$

$$+ 3\mathbf{K}_{xy} \circ \mathbf{K}_{xy}\left\langle \frac{\mathbf{K}_{xy}}{N^2}\right\rangle + 2\mathbf{K}_{xy} \circ \mathbf{H}\mathbf{K}_{xy} \circ \mathbf{H}\mathbf{K}_{xy}$$

$$+ 2\mathbf{K}_{xy} \circ \mathbf{K}_{xy}\mathbf{H} \circ \mathbf{K}_{xy}\mathbf{H} - 6\mathbf{K}_{xy} \circ \mathbf{K}_{xy}\mathbf{H}\left\langle \frac{\mathbf{K}_{xy}}{N^2}\right\rangle$$

$$- 6\mathbf{K}_{xy} \circ \mathbf{H}\mathbf{K}_{xy}\left\langle \frac{\mathbf{K}_{xy}}{N^2}\right\rangle + 4\left\langle \frac{\mathbf{K}}{N^2}\right\rangle^2 \mathbf{K}_{xy}\bigg\rangle.$$

We mention also that the first two terms $\|\kappa_k^{(3)}(\gamma)\|_{\mathcal{H}_k^{\otimes 3}}^2, \|\kappa_k^{(3)}(\eta)\|_{\mathcal{H}_k^{\otimes 3}}^2$ can be computed a little more simply than in Example E.2 since the expressions have more symmetry, using the notation $\mathbf{K}_x = [k(x_i,x_j)]_{i,j=1}^N$ we can write down the V-statistic for $\|\kappa_k^{(3)}(\gamma)\|_{\mathcal{H}_k^{\otimes 3}}^2$ as

$$\frac{1}{N^2}\bigg\langle \mathbf{K}_x \circ \mathbf{K}_x \circ \mathbf{K}_x - 6\mathbf{K}_x \circ \mathbf{K}_x\mathbf{H} \circ \mathbf{K}_x$$

$$+ 4\mathbf{K}_x\mathbf{H} \circ \mathbf{K}_x \circ \mathbf{K}_x\mathbf{H} + 3\mathbf{K}_x \circ \mathbf{K}_x\left\langle \frac{\mathbf{K}_x}{N^2}\right\rangle$$

$$+ 6\mathbf{K}_x\mathbf{H} \circ \mathbf{H}\mathbf{K}_x \circ \mathbf{K}_x - 12\mathbf{K}_x \circ \mathbf{H}\mathbf{K}_x\left\langle \frac{\mathbf{K}_x}{N^2}\right\rangle$$

$$+ 4\left\langle \frac{\mathbf{K}_x}{N^2}\right\rangle^2 \mathbf{K}_x\bigg\rangle$$

with a similar expression for $\|\kappa_k^{(3)}(\eta)\|_{\mathcal{H}_k^{\otimes 3}}^2$. The estimator can be computed in quadratic time.

