# OpenReview forum: "Kernelized Cumulants: Beyond Kernel Mean Embeddings"
_NeurIPS.cc/2023/Conference — NeurIPS 2023 spotlight_

### Official Review · Reviewer_6dmY · 2023-07-03

**Soundness:** 3 good
**Presentation:** 3 good
**Contribution:** 2 fair
**Rating:** 6
**Confidence:** 3

**Summary:**

This paper proposes kernelized cumulants to extend classical cumulants in $\mathbb{R}^d$ and shows that the kernelized cumulants provide a new set of all-purpose statistics and are computationally tractable.
This paper also show advantages of kernelized cumulants both theoretically and empirically.

**Strengths:**

* The paper is well-written and easy to follow.

* The method, that is, kernelized cumulant, proposed in this paper is novel and might be useful in real-world applications.
For example, this method can be used to provide metrics for distributions and measure independence.
Moreover, this method includes traditional MMD and HSIC as its special case and can even outperform the traditional ones.


**Weaknesses:**

* In the experiments, more kernels, such as the neural tangent kernel, could be considered.
* It would be better if the authors provide theoretical guarantees of the effectiveness of kernelized cumulants, such as consistency.

**Questions:**

1. In the experiments, what is the criteria of the 'optimal value' of $\sigma$ (Line 265)?

**Limitations:**

The authors have adequately addressed the limitations.

---

> ### Author Rebuttal · Authors · 2023-08-08
>
> We thank the reviewer for the time and effort invested, and for the kind review. Below, we answer the questions in detail.
>
>  - ”more kernels, such as the neural tangent kernel, could be considered.”
> Yes, essentially any kernel can be used and in our experiments we focused on the standard kernel choices by practitioners.
> We are not sure what the reviewer exactly means by using the neural tangent kernel in this context; it would be certainly interesting to
> use the presented kernelized cumulants to study how dependencies arise in the training of neural networks, although this
> is beyond the scope of the current work.
>
>  - ”provide theoretical guarantees”
> This is a good point, as the estimators are V-statistics they inherit all of their nice properties like consistency, so this boils
> down to citing the standard references for V-statistics. We will clarify this in the main text.
>
> - ”what is the criteria of the ’optimal value’ of $\sigma$?”
> By ’optimal value’ we mean the value for which the test has the highest power; we will clarify this in the main text.
>
> We hope that our answers resolve all the questions.

---

> > ### Comment · Reviewer_6dmY · 2023-08-21
> >
> > Thanks for the response, but I will keep my score.

---

### Official Review · Reviewer_h8uP · 2023-07-05

**Soundness:** 3 good
**Presentation:** 3 good
**Contribution:** 3 good
**Rating:** 7
**Confidence:** 3

**Summary:**

This paper generalizes the notion of cumulants to Hilbert-space-valued random variables.
When these Hilbert spaces are RKHSs, the kernel trick applies so that computations can be performed with the kernel function.
It leads to higher-order two-sample and independence tests, which generalizes MMD and HSIC.
The efficiency of these tests is demonstrated numerically on both synthetic and real data.

EDIT: I have read the author's rebuttal, which partially addressed my concerns.

**Strengths:**

The paper is well written and clear, despite the fact that the notation is heavy due to the complexity of the considered objects.
I appreciated that equivalent definitions of the kernelized cumulants are given, as well as special cases to build intuition and relate it to classical notions.

Because of this, I think the paper should be accepted.

**Weaknesses:**

A first weakness in my opinion is that the main contribution of the paper is a straightforward combination of two known concepts: (i) cumulants and (ii) the use of the kernel methods in statistical testing. Also, rather than using the tensor algebra framework used by the authors in Section 3 and Appendix C.2, it seems to me that the simpler route would be to use the generating function $K(\theta_1, \dots, \theta_d) = \log \mathbb E[ \exp(\sum_{i=1}^d \langle \theta_i, X_i \rangle ) ]$ and define the cumulants from its series expansion, or equivalently derivatives at zero (which would be the correct equivalent of item 1. in Appendix C.1, rather than eq. (11)) . The definitions are of course equivalent, but it would lighten the formalism in the main text.

The second main weakness is that the relationship between MMD/HSIC and the higher-order versions *when one is allowed to change the kernel* is not discussed. If I'm not mistaken, the second-moment embedding of a probability distribution with kernel $k$ coincides with the mean embedding with the squared kernel $k^2$. So one should expect a relationship between $d^{(2)}$ with kernel $k$ and MMD with some combination of $k$ and $k^2$ as kernel. If that is the case, then the use of higher-order cumulants can be equivalently rephrased as the use of different kernels. In particular, it is not clear whether one could be better off by sticking to MMD/HSIC but with well-designed kernels, as the estimation of higher-order moments has a higher variance. I think this point should be discussed in the text.

Two additional minor remarks:
- Repetition of "energy distance" in lines 37-38, which I'm guessing is a typo.
- Missing related work: [1] considers a second-moment kernel embedding and defines kernel information-theoretic quantities.

[1] Bach, Francis. "Information theory with kernel methods." IEEE Transactions on Information Theory 69.2 (2022): 752-775.

**Questions:**

- I am suspicious of the fact that the computational complexity of MMD/HSIC and the higher-order extensions are the same, and are bottlenecked by the computation of the kernel matrix. Surely the complexity must increase with $m = \mathrm{deg}(\mathbf{i})$, even if it remains quadratic in the sample size?

- Is there a relationship between going to higher-order cumulants and changing the kernel? Can we achieve the same performance than the higher-order cumulants by adapting the kernel?

- Isn't the the $V$ statistic estimator a straightforward replacement of the expectation with empirical averages over the sample? I think this could be mentioned in the text.

**Limitations:**

The authors mentioned the two main limitations of their approach:
- It is not clear how to design the kernels to maximize the performance of the statistical tests.
- There is no theoretical analysis of the introduced tests.

I agree that a complete resolution of these issues should be left to future work. However, I think the second weakness above, which is related to the first limitation, should be at least acknowledged.

There is no foreseeable negative societal impact.

---

> ### Author Rebuttal · Authors · 2023-08-08
>
> We thank the reviewer for the time and effort invested, and for the kind review. Below, we answer the questions in detail.
>
>  - ”use the generating function”
> We agree that this is a more intuitive way to introduce the concept, but it is not very instructive when it comes to
> actually computing the statistics or writing down estimators, and in order to do so it is typical to work with combinatorial
> descriptions of cumulants. In order to save space we introduced only the combinatorial one as it was necessary to derive
> the statistics in the later parts of the paper.
>
>  - ”higher-order cumulants can be equivalently rephrased as the use of different kernels”
> This is true for higher order moments, but we stress that the cumulants are not equivalent to mean embeddings of
> different kernels. To illustrate this, the embedding for the second cumulant of a random variable $X$ is $ \mathbb{E}\big[k(X,\cdot)\otimes k(X,\cdot)\big] - \mathbb{E}\big[k(X,\cdot)\otimes k(Y,\cdot)\big] $ where $Y$ is an i.i.d copy of $X$. The first term here is indeed a mean embedding of the product kernel
> $k^{\otimes 2}$, but the second term acts on a different product measure than the first. This structured cancellation of the various
> moments of the measure is central to the cumulants ability to pick up the higher order features, and makes them distinct
> from ”only” using higher order moments of kernels.
>
>  - ”Surely the complexity must increase with the degree”
> This is true, we apologize that this was not made sufficiently clear in the text. The complexity in the sample size is
> quadratic for all the proposed statistics, but the number of operations to compute them does increase with the degree and
> is determined by the number of partitions corresponding to that degree. The number of partitions does grow quickly as
> the degree gets large, but for reasonably sized degrees it is not an issue. See also our reply to R1 on this topic, particularly
> the paragraph starting with ’Degree dependence’.
>
>  - ”Isn’t the the V-statistic estimator a straightforward replacement of the expectation”
> It is very similar but there is some subtlety in which indices are summed over when computing the V-statistics, as an
> example consider the two generalised moments of a random variable $X$, $\mathbb{E}(X^2)$ and $[\mathbb{E}(X)]^2$. Given $N$ samples $x_1, \ldots, x_N$ the V-statistic for the first one is $\frac{1}{N}\sum_{i=1}^N x_ix_i$ and the V-statistic for the other is $\frac{1}{N^2}\sum_{i=1}^N\sum_{j=1}^N x_ix_j$.
>
>  - Minor remarks:
> Thank you, we will address this and add the relevant citation. Regarding the reference, (i) it falls under the umbrella of
> $k^{\otimes 2}$-based information theoretical quantities, (ii) it relies on the uncentered covariance operator, (iii) it requires the kernel
> to be universal (in contrast to our significantly relaxed point-separating assumption), and (iv) its complexity (justified in
> the submission for the case of kernel entropy) is high, cubic in the sample size (whereas our estimator for fixed degree is
> quadratic in the sample size). We are happy to include a citation to the mentioned paper as a related work.
>
> We hope that our answers resolve all the questions.

---

> > ### Comment · Reviewer_h8uP · 2023-08-11
> >
> > Thank you for your detailed answer.
> >
> > - I agree the generating function is not useful in practice. I understand the choice of skipping it, though it could be mentioned in the appendix.
> > - Your second point is important. In order to see the second cumulant as a mean embedding with a different kernel, we thus need to consider pairs by replacing the random variable $X$ to $(X, X')$ where $X'$ is an iid copy of $X$, and then use the kernel $k((x,x'),(y,y')) = k(x,y)^2 - k(x,y)k(x',y) - k(x,y)k(x',y) + k(x,y')k(x',y)$ (among other possibilities). I think this equivalence between higher-order cumulants and kernels over tuples of data points, to be contrasted with the equivalence between higher-order moments and product kernels, is important to understand the expressivity of higher-order cumulants.
> >
> > I hope that this discussion will help the authors improve the clarity of the paper concerning these more technical points. I recommend acceptance.

---

### Official Review · Reviewer_952h · 2023-07-06

**Soundness:** 3 good
**Presentation:** 3 good
**Contribution:** 3 good
**Rating:** 7
**Confidence:** 3

**Summary:**

The authors introduce the kernelized cumulant and show that it can characterize distributions and statistical (in)dependence.

**Strengths:**

1. The kernelized cumulant provides a natural generalization of the popular maximum mean discrepancy (MMD) as well as the Hilbert-Schmidt independence criterion (HSIC).
2. The authors illustrate several interesting properties of the kernelized cumulants and provide a two-sample test for non-characteristic feature maps as well as an independence test.
3. The authors demonstrate the utility of the proposed method on a variety of datasets with competitive results.

**Weaknesses:**

1. Estimating the kernelized cumulant may require a large sample. Convergence rate of the estimator is not discussed, but the empirical results suggest comparable performance as the HSIC.
2. The proposed method could potentially be vulnerable to kernel misspecification.

**Questions:**

1. Is the independence criterion kernel dependent? What happens if the kernel is misspecified?
2. What are the convergence rates for the finite-sample estimators?

---

> ### Author Rebuttal · Authors · 2023-08-08
>
> We thank the reviewer for the time and effort invested, and for the kind review. Below, we answer the questions in detail.
>
>  - ”Is the independence criterion kernel dependent?”
> Since we are working in a very general framework, the independence criterion works for a very wide family of kernels,
> even for simple ones such as the linear kernel on $\mathbb{R}^d$. However, our proposed methods suffer from the same drawbacks
> as independence testing with HSIC or 2-sample testing with MMD in that the testing power can be severely affected by
> picking an inappropriate kernel.
>
>  - ”Convergence rates”
> As the estimator of our proposed kernelized cumulant is a sum of V-statistics, the rates are inherited from that of V-
> statistics.
>
>  - ”kernel misspecification”
> In our analysis we imposed minimal assumptions on the kernel (point-separating) to allow for the wide applicability of
> the proposed framework. Understanding the optimal choice of kernels (we assume that this is what the reviewer means
> by kernel misspecification) or even the involved hyperparameters specifically for MMD (which corresponds to a degree one
> object in our work) in specific downstream tasks is an important and highly non-trivial problem. Probably the simplest
> task in this context is 2-sample testing, where one can phrase the goal to achieve minimax optimality. In this very specific
> case, the analysis for constructing (almost) minimax optimal MMD-based adaptive tests can be worked out in 63 pages
> [7]. Extension of their results could mean the first step of the grounding of kernelized cumulants in downstream tasks.
> Similarly, getting optimal rates even for the classic MMD and for radial universal kernel is non-straightforward; to our best
> knowledge, the only available result in this domain is [21]. Our investigated assumptions are weaker (hence the analysis is
> expected to be harder) in multiple aspects: the space is Polish (hence even the notion of radial property is not defined),
> cumulants need not be 1st degree objects (as MMD), and our results (Theorem 2-3) hold for point-separating kernels
> (which requirement is significantly weaker than characteristic property, which itself is a specific case of universality).
> We agree (in line with the penultimate paragraph of Section 1) that these are important future research directions.
>
> We hope that our answers resolve all the questions.

---

> > ### Comment · Reviewer_952h · 2023-08-18
> > **After rebuttal**
> >
> > Thank you for addressing my comments. I've increased my score accordingly.

---

### Official Review · Reviewer_Lvaf · 2023-07-06

**Soundness:** 3 good
**Presentation:** 3 good
**Contribution:** 4 excellent
**Rating:** 7
**Confidence:** 3

**Summary:**

This papers revisits advances in cumulants on real data, by extending them to provide cumulants for random variables in an RKHS.

**Strengths:**

The idea of this work is interesting, as the paper proposed to go beyond conventional kernel mean and kernel covariance. Moreover, a proposed kernel trick allow to obtain efficiently the kernelized cumulants.
The paper well describes the contributions. The derivations seem to be good, and some experiments on synthetic and real datasets allow to understand the relevance of the proposed approach

**Weaknesses:**

While the paper includes some experimental results, there are only two settings : independence test and two-sample test (MMD-like). Is there any other  tests (or beyond test) where the proposed kernelized cumulants would be relevant ?

**Questions:**

see question in weaknesses

**Limitations:**

OK

---

> ### Author Rebuttal · Authors · 2023-08-08
>
> We thank the reviewer for the time and effort invested, and for the kind review.
>
>  - ”Is there any other tests (or beyond test) where the proposed kernelized cumulants would be relevant?”
> The proposed kernelized-cumulant based measures are general divergence and independence measures, hence can be used
> practically in any application which rely on information theoretical objective and expectedly with improved sample-
> efficiency (as our hypothesis testing illustrations in the submission suggest). Examples include feature selection [2, 18, 9, 22],
> causal discovery [14, 13, 16, 3, 17], distribution classification [11, 23] and regression [20, 19, 8, 6, 15], or generative adversarial
> networks [5, 10, 1].
>
> We hope that our answer resolves the reviewer’s question.

---

> > ### Comment · Reviewer_Lvaf · 2023-08-14
> >
> > I would like to thank the authors for taking the time to respond to the raised question.  Considering the issues brought up by myself and the other reviewers, as well as the rebuttal, I am maintaining my "accept" score.

---

### Official Review · Reviewer_Q1mu · 2023-07-12

**Soundness:** 4 excellent
**Presentation:** 3 good
**Contribution:** 3 good
**Rating:** 7
**Confidence:** 4

**Summary:**

The paper generalises the notion of kernel mean embeddings to higher-order cumulants. It proposes kernelled cumulants in the RKHS. While kernelled cumulants reside in tensor product space of the RKHS, the paper shows that Hilbert space metric between cumulants can be exactly computed using the kernel trick. Based on this construction, the paper proposes: (1) a two-sample test statistics, that generalises MMD test statistics by considering distance between cumulants, and (2) a generalisation of HSIC statistic for independence testing, again by considering distant between cumulants of joint distribution and product of marginals.
The advantages of the construction and proposed tests include: (i) the test are applicable for a broader class of "point-separating" kernels (unlike MMD/HSIC that can only useful for characteristic kernels); (ii) the new statistics can be computed in quadratic time (same as MMD); and (iii) empirically achieve higher power that classical MMD/HSIC statistics (both on synthetic and real data).

**Strengths:**

- The idea of considering higher order moments/cumulants in RKHS is quite natural, and yet unexplored in the literature (apart from the recent work of Makigusa (2020) that consider only 2nd order moments). Hence, the contribution is novel and quite timely
- The main strength of the paper is that the computational cost of the proposed statistics is still quadratic in sample size (same as MMD), which implies the advantages of higher order moments does not come at significant additional cost
- The work is technically sound, and the construction of cumulant and use of kernel trick is reasonably involved.

**Weaknesses:**

- While it is easy to imagine in general that higher-order cumulants can distinguish between more distributions, the advantage of kernelled cumulants is difficult to grasp. If a characteristic kernel is used wouldn't mean embedding (MMD) suffice?
- The line of argument used in the paper to demonstrate advantage of kernel cumulants is that (empirically) they have show higher power (can detect small differences better). Is there a theoretical justification for this? It would be sufficient if the authors provide a justification / reference that standard (non-kernel) cumulants are more sample efficient in some cases, where means already show separation

-  For the synthetic experiments, the null rejection rate should also be plotted to show that cumulants do not have higher tendency to reject than MMD/HSIC. While this is true for the real data, unfortunately both 1st order and higher order terms reject at a rate higher than significance level

- Overall, it is not clear when tests based on higher-order cumulants are indeed needed in practice. I still feel the work is relevant, but some discussions on this would certainly increase the significance of the work for the broader community
- The paper, although well-written, is quite dense and at times bit difficult to follow, but this can be attributed to the content of the paper

**Questions:**

- see weakness
- in addition, a precise statement on computational complexity (at least for d2, d3) would be useful

Minor remarks: H in Lemma 3 is not defined in main paper (but in appendix), and E et al citation seems incorrect (there is a full name for E)

**Limitations:**

the paper does not have immediate negative societal impact (although conclusions from hypothesis tests can always have). Hence, the work could benefit from:
- consistency results (similar to kernel two-sample tests)
- characterisation of whether higher-order cumulant based statistics typically tend to be larger than MMD (even under null). The comment is about sample estimates and not expected value (hence, more tied to concentration/consistency of test statistics)

---

> ### Author Rebuttal · Authors · 2023-08-08
>
> We thank the reviewer for the time and effort invested, and for the kind review. Below, we answer the questions in detail.
>
>  - ”If a characteristic kernel is used wouldn’t mean embedding (MMD) suffice?”
> This is true of course, a characteristic kernel is theoretically sufficient for 2-sample testing (in MMD), and a universal one
> is sufficient for independence testing (in HSIC). The main use case of cumulants are for when either one is in a situation
> where a non-characteristic kernel is preferable, or one does have a characteristic kernel but the data is structured in a way
> where one gets additional test power from using the cumulants and hence estimators which are more sample-efficient, like
> in the experiments shown in the main text.
>
>  - ”Is there a theoretical justification for the higher testing power?”
> Testing power for these kinds of tests do not normally come with satisfactory descriptions since the tests are infinite dimensional. Closed form expressions of power are usually expressed as infinite convergent series, see for example Theorem 3.1 [16]. because of this, trying to study theoretical properties of testing power in this setting is a very challenging problem, and while an interesting one, we believe that the experimental results are convincing, and that further theoretical study is beyond the scope of the article.
>
>  - ”both 1st order and higher order terms reject at a rate higher than significance level”
> It is true that the rejection rate is higher than 5%, but as it is consistent for both methods we do not feel that it gives any
> unfair advantages either way. The reason for this is most likely due to the way we choose the hyperparameter by optimising
> for test power, which would incur a bias for the rejection rate. We did include rejection rates for the bicycle data in Figure
> 8 in the appendix where it can be seen that it is around 8% for both methods, and we can include the rejection rates for
> the synthetics experiments here too.
>
>  - ”(. . . ) when tests based on higher-order cumulants are indeed needed in practice. I still feel the work is relevant, but some
> discussions on this would certainly increase the significance of the work for the broader community”
> One answer is that kernelized cumulants provide guarantees even when non-characteristic kernels are used. However, even for characteristic kernels the presented kernelized cumulants are advantageous since they lead to more sample-efficient estimators and hence tests. To see this on the simplest ”unkernelized” example would be to test if two normal distributed random variables
> have the same distribution. Clearly, here the best answer is to just compute the sample means and sample covariances
> and then reject/accept if they are close enough; in contrast computing mean and second moments would lead to much
> more noisy tests and more samples would be needed (see the discussion in Appendix A). The same arguments carry over
> when we use kernels for testing. Our experiments confirm this intuition and show that the tests given by statistics derived
> from kernelized cumulants are usually more efficient than the classic kernelized statistics. Hence, kernelized cumulants
> are usually preferrable. However, the cost of the somewhat more complicated test statistics get harder to implement, the
> higher m becomes. On a practical side, we believe using m = 2 is a good compromise between gain in sample efficiency
> (hence better tests) and ease of implementation. We will add a sentence to emphasize this.
>
>  - ”A precise statement on computational complexity”
>
> **Sample size dependence:** The exact statistic used are expanded on in examples E.1, E.2, and E.3 in the appendix. Since
> all these statistics can be phrased as polynomials of Gram matrices and some multiplication with centering matrices, the
> most expensive computation is computing the Gram matrices themselves. For example, when computing $d^{(2)}$, assume that
> one has N samples from one measure and M from the other. One first computes the 3 relevant Gram matrices which is
> $O(N^2 +NM + M^2)$ and depends on the specific kernel used. After that one computes the centered matrices which involves
> $N^2 + 2NM + M^2$ additions and the same number of divisions. One then computes the Hadamard square of the centered
> matrices which involves $N^2 +NM + M^2$ multiplications and finally sums over all the elements and normalizes the output
> which involves a further $N^2 +NM + M^2$ additions and 3 divisions, and then 3 more additions to add them up. Taking the
> total number of operations (in addition to computing the Gram matrices), this means $2N^2 + 2M^2 + 3NM + 3$ additions
> and the same number of multiplications/divisions.
>
> **Degree dependence:** There is no free lunch. The $m$-th Bell number $B_m$ (https://en.wikipedia.org/wiki/Bell_number)
> is defined as the number of elements in $P (m)$. The Bell numbers follow a recursion: $B_{m+1} = |P(m+1)|  = \sum_{k=0}^m \binom{m}{k} B_{k}$, with  the first elements of the sequence being $B_0 = B_1 = 1$, $B_2 = 2$, $B_3 = 5$, $B_4 = 15$, $B_5 = 52$, $B_6 = 203$, $B_7=877$, $B_8=4140$. By (6)-(7), in the worst case the number of operations to compute $d^{(i)}(\gamma,\eta)$  or $\Vert\kappa^{i}_{k_1,\ldots,k_d}(\gamma)\Vert^2$ is proportional to $B_m^2$ (it equals to $3B_m^2$ and to $B_m^2$, respectively). It is true that asymptotically $B_m$ gets very large [4, 12], but
> for reasonably small degrees the computation is still manageable. In addition, merging various terms in the estimator can
> often be carried out, which leads to computational saving. For instance, the estimator of $d^{(2)}$ (see Lemma 2, Example E.1),
> CSIC (Lemma 3, Example E.2) and $d^{(3)}$ (Example E.3) consists of only $2$, $11$ and  $10 + 2\times 7 = 24$ terms compared to the
> predicted worst-case setting of $3B_2^2 =12$, $B_3^2 = 25$, and $3B_3^2=75$ terms, respectively.
>
>  - Minor remarks:
> Thank you for pointing these out, we will address them.
>
> We hope that our answers resolve all the questions.

---

> > ### Comment · Reviewer_Q1mu · 2023-08-10
> > **Thank you**
> >
> > I thank the authors for the responses, and clarifying the relevance of looking at higher order cumulants. In future, it would help to further investigate practical implications of kernel cumulants. It feels like there is more potential than what is claimed in the paper/rebuttal.

---

### Author Rebuttal · Authors · 2023-08-08

**References for the rebuttal:**

[1] Mikolaj Binkowski, Danica Sutherland, Michael Arbel, and Arthur Gretton. Demystifying MMD GANs. In International
Conference on Learning Representations (ICLR), 2018.

[2] Gustavo Camps-Valls, Joris M. Mooij, and Bernhard Sch ̈olkopf. Remote sensing feature selection by kernel dependence
measures. IEEE Geoscience and Remote Sensing Letters, 7(3):587–591, 2010.

[3] Shubhadeep Chakraborty and Xianyang Zhang. Distance metrics for measuring joint dependence with application to causal
inference. Journal of the American Statistical Association, 114(528):1638–1650, 2019.

[4] N. G. de Bruijn. Asymptotic Methods in Analysis. Dover, 1981.

[5] Gintare Karolina Dziugaite, Daniel M. Roy, and Zoubin Ghahramani. Training generative neural networks via maximum
mean discrepancy optimization. In Conference on Uncertainty in Artificial Intelligence (UAI), pages 258–267, 2015.

[6] Zhiying Fang, Zheng-Chu Guo, and Ding-Xuan Zhou. Optimal learning rates for distribution regression. Journal of
Complexity, page 101426, 2020.

[7] Omar Hagrass, Bharath K. Sriperumbudur, and Bing Li. Spectral regularized kernel two-sample tests. Technical report,
2022. (https://arxiv.org/abs/2212.09201).

[8] Ho Chung Leon Law, Danica Sutherland, Dino Sejdinovic, and Seth Flaxman. Bayesian approaches to distribution regression.
International Conference on Artificial Intelligence and Statistics (AISTATS), 84:1167–1176, 2018.

[9] Runze Li, Wei Zhong, and Liping Zhu. Feature screening via distance correlation learning. Journal of the American
Statistical Association, 107(499):1129–1139, 2012.

[10] Yujia Li, Kevin Swersky, and Richard Zemel. Generative moment matching networks. In International Conference on
Machine Learning (ICML), pages 1718–1727, 2015.

[11] David Lopez-Paz, Krikamol Muandet, Bernhard Schölkopf, and Iliya Tolstikhin. Towards a learning theory of cause-effect
inference. International Conference on Machine Learning (ICML), 37:1452–1461, 2015.

[12] Laszlo Lovasz. Combinatorial Problems and Exercise. 2nd ed. Amsterdam, Netherlands: North-Holland, 1993.

[13] Joris Mooij, Jonas Peters, Dominik Janzing, Jakob Zscheischler, and Bernhard Schölkopf. Distinguishing cause from effect
using observational data: Methods and benchmarks. Journal of Machine Learning Research, 17:1–102, 2016.

[14] Krikamol Muandet, Kenji Fukumizu, Francesco Dinuzzo, and Bernhard Schölkopf. Learning from distributions via support
measure machines. In Advances in Neural Information Processing Systems (NIPS), pages 10–18, 2011.

[15] Nicole Mucke. Stochastic gradient descent meets distribution regression. In International Conference on Artificial Intelli-
gence and Statistics (AISTATS), pages 2143–2151, 2021.

[16] Niklas Pfister, Peter Buhlmann, Bernhard Schölkopf, and Jonas Peters. Kernel-based tests for joint independence. Journal
of the Royal Statistical Society: Series B (Statistical Methodology), 80(1):5–31, 2018.

[17] Bernhard Schölkopf, Francesco Locatello, Stefan Bauer, Nan Rosemary Ke, Nal Kalchbrenner, Anirudh Goyal, and Yoshua
Bengio. Toward causal representation learning. Proceedings of the IEEE, 109(5):612–634, 2021.

[18] Le Song, Alex Smola, Arthur Gretton, Justin Bedo, and Karsten Borgwardt. Feature selection via dependence maximization.
Journal of Machine Learning Research, 13(1):1393–1434, 2012.

[19] Danica Sutherland, Junier Oliva, Barnabas Poczos, and Jeff Schneider. Linear-time learning on distributions with approxi-
mate kernel embeddings. In AAAI Conference on Artifical Intelligence (AAAI), pages 2073–2079, 2016.

[20] Zoltan Szabo, Bharath K. Sriperumbudur, Barnab ́as P ́oczos, and Arthur Gretton. Learning theory for distribution regression.
Journal of Machine Learning Research, 17(152):1–40, 2016.

[21] Ilya Tolstikhin, Bharath Sriperumbudur, and Bernhard Schölkopf. Minimax estimation of maximal mean discrepancy with
radial kernels. In Advances in Neural Information Processing Systems (NIPS), pages 1930–1938, 2016.

[22] Andi Wang, Juan Du, Xi Zhang, and Jianjun Shi. Ranking features to promote diversity: An approach based on sparse
distance correlation. Technometrics, 64(3):384–395, 2022.

[23] Manzil Zaheer, Satwik Kottur, Siamak Ravanbakhsh, Barnab ́as P ́oczos, Ruslan Salakhutdinov, and Alexander Smola. Deep
sets. In Advances in Neural Information Processing Systems (NIPS), pages 3394–3404, 2017.

---

### Decision · Program_Chairs · 2023-09-21

**Decision:**

Accept (spotlight)

**Comment:**

The reviewers unanimously recommended to accept the paper.